



# 1 Hydrology and Water Resources Management in Ancient
# 2 India

Pushpendra Kumar Singh[1], Pankaj Dey[2], Sharad Kumar Jain[3] and Pradeep Mujumdar[2,4]
[1] Water Resources Systems Division, National Institute of Hydrology, Roorkee, 247667, India
[2] Department of Civil Engineering, Indian Institute of Science, Bangalore, 560012, India
[3] Visiting Professor, Department of Civil Engineering, Indian Institute of Technology, Roorkee, 247667, India
[4] Interdisciplinary Centre for Water Research, Indian Institute of Science, Bangalore, 560012, India
*Correspondence to*: P P Mujumdar (pradeep@iisc.ac.in)
**Abstract.** Hydrologic knowledge in India has a historical footprint extending over several millenniums through
the Harappan Civilisation (~ 3000 BC – 1500 BC) and the Vedic period (~1500-500 BC). As in other ancient
civilisations across the world, the need to manage water propelled the growth of hydrologic science in ancient
India also. Most of the ancient hydrologic knowledge, however, has remained hidden and unexplored to the world
at large till the recent times. In this paper, we provide some fascinating glimpses into the hydrological, hydraulic
and related engineering knowledge that existed in ancient India, as discussed in contemporary literature and in the
recent explorations and findings. The Vedas, particularly, the *Rigveda*, *Yajurveda* and *Atharvaveda* have many
references to water cycle and associated processes, including water quality, hydraulic machines and other
structures and nature-based solutions (NBS) for water management. The Harappan Civilization epitomizes the
level of development of water sciences in ancient India that includes construction of sophisticated hydraulic
structures, wastewater disposal systems based on centralized and decentralized concepts as well as methods for
wastewater treatments. The Mauryan empire (~ 322 BC – 185 BC) is credited as the first "hydraulic civilization"
characterised by construction of dams with spillways, reservoirs, channels equipped with spillways, pynes and
*Ahars*, understanding of water balance, development of water pricing systems, measurement of rainfall and
knowledge of the various hydrological processes. As we investigate deeper into hydrologic references in ancient
literature, including the Indian mythology, many fascinating dimensions of the early scientific endeavours of
Indians emerge.
**1 Introduction**
Water is intimately linked to human existence and is the source of societal and cultural development, traditions,
rituals and religious beliefs. The humans created permanent settlements about 10,000 years ago when they adopted
an agrarian way of life and started developing different socio-cultural societies and settlements, largely dependent
on water in one way or other (Vuorinen et al., 2007). These developments established a unique relationship
between humans and water. Most of the ancient civilizations, e.g., the Indus Valley, Egyptian, Mesopotamian,
and Chinese civilizations were developed at places where water required for agricultural and human needs was
readily available, i.e., in the vicinity of springs, lakes, rivers and low sea levels (Yannopoulos et al., 2015). As



water was the prime mover of the ancient civilizations, a clear understanding of the hydrologic cycle, nature and
pattern of its various components along with water uses for different purposes led these civilizations to flourish
for thousands of years.

The Harappan (or Indus Valley) civilization (3000 BC – 1500 BC), one of the earliest and most advanced
civilizations, was also the world's largest in spatial extent and epitomises the level of development of science and
society in proto-historic Indian sub-continent. Jansen (1989) states that the citizens of Harappan civilization were
known for their obsession with water; they prayed to the rivers every day and accorded the rivers a divine status.
The urban centres were developed with state-of-the art civil and architectural designs with provisions of
sophisticated drainage and waste water management systems. Agriculture was the main economic activity of the
society and an extensive network of reservoirs, wells, canals along with low cost water harvesting techniques were
developed throughout the region at that time (Nair, 2004). The Mohenjo-Daro and Dholavira, major cities of Indus
Valley are the best examples having the state-of-the art water management and drainage systems. The Great Bath
of Mohenjo-Daro of Indus Valley is considered as the "earliest public water tank of the ancient world". The
"*Arthashastra*" attributed to Kautilya "who reportedly was the chief minister to the emperor Chandragupta (300
BC), the founder of the Mauryan dynasty" (Encyclopaedia Britannica, https://www.britannica.com/topic/Artha-
shastra) deals with several issues of governance, including water governance. It mentions about a manually
operated cooling device "Variyantra" (revolving water spray for cooling the air).   It also gives an extensive
account of hydraulic structures built for irrigation and other purposes during the period of the Mauryan empire
(Shamasastry, 1961).

The Pynes and *Ahars* (combined irrigation and water management system), reservoir (Sudarshan lake) at Girnar
and many other structures were also built during the Mauryan empire (322-185 BC). McClellan III and Dorn,
(2015) noted that '… the Mauryan empire was first and foremost a great hydraulic civilization …'. This reflects
that the technology of the construction of the dams, reservoirs, channels, measurement of rainfall and knowledge
of the various hydrological process was well known to the ancient Indian society. The water pricing was also an
important component of the water management system in Mauryan empire. There are also adequate archaeological
evidences to testify that the Harappans of the Indus Valley were well aware of the seasonal rainfall and flooding
of the river Indus during the period between 2500 and 1700 B.C., which is corroborated by modern meteorological
investigations (Srinivasan, 1976).

The Vedic texts, which were composed probably between 1500 and 1200 BC (1700–1100 BC according to some
scholars), contain valuable references to 'hydrological cycle'. It was known during Vedic and later times
(Rigveda, VIII, 6.19, VIII, 6.20; and VIII, 12.3) (Sarasvati, 2009) that water is not lost in the various processes of
hydrological cycle namely evaporation, condensation, rainfall, streamflow, etc., but gets converted from one form
to another. Indians were, at that time, acquainted with cyclonic and orographic effects on rainfall (*Vayu Purana*)
and radiation, and convectional heating of earth and evapotranspiration. The Vedic texts and other Mauryan period
texts such as '*Arthshastra*' mention about other hydrologic processes such as infiltration, interception, streamflow
and geomorphology, including the erosion process. Reference to the hydrologic cycle and artesian wells is
available in *Ramayana* (200 B.C.) (Vālmīki and Goswami, 1973). Ground water development and water quality



considerations also received sufficient attention in ancient India, as evident from the *Brihat Samhita* (550 A.D.)
(Jha, 1988). Topics such as water uptake by plants, evaporation, clouds and their characteristics along with rainfall
prediction by observing the natural phenomena of previous years, had been discussed in *Brihat Samhita* (550
A.D.), *Meghamala* (900 A.D.) and other literature from ancient India.

Historical development of hydro-science has been dealt by many researchers (Baker and Horton, 1936; Biswas,
1969; Chow, 1964). However, not many references to the hydrological contributions in ancient India are found.
Chow (1974) rightly mentions that "… the history of hydrology in Asia is fragmentary at best and much insight
could be obtained by further study". According to Mujumdar and Jain (2018), there is rigorous discussion in
ancient Indian literature on several aspects of hydrologic processes and water resources development and
management practices as we understand them today.

Evidences from ancient water history provide an insight into the hydrological knowledge generated by Indians
more than 3000 years ago. This paper explores the many facets of ancient Indian knowledge on hydrology and
water resources with focus on various hydrological processes, water management and technology, and wastewater
management, based on earlier reviews of Indian scriptures such as the Vedas, the *Arthasastra* (Shamasastry,
1961), *Astadhyayi* (Jigyasu, 1979), *Ramayana* (Vālmīki and Goswami, 1973), *Mahabharata*, *Puranas*, *Brihat*
*Samhita* (Jha, 1988), *Meghmala*, *Mayurchitraka*, Jain and Buddhist texts and other ancient texts.
**2 Knowledge of Hydrological Processes in Ancient India**
Hydrologic cycle is the most fundamental concept in hydrology that involves the total earth system comprising
the atmosphere (the gaseous envelop), the hydrosphere (surface and subsurface water), lithosphere (soils and
rocks), the biosphere (plants and animals), and the Oceans. Water passes through these five spheres of the earth
system in one or more of the three phases: solid (ice), liquid and vapour. The *Rigveda*, which is an ancient religious
scripture, contains many references to hydrologic cycle and associated processes (Sarasvati, 2009). The *Rigveda*
mentions that 'the God has created Sun and placed it in such a position that it illuminates the whole universe and
extracts water continuously (in the form of vapour) and then converts it to cloud and ultimately discharges as rain'
(Verse, I, 7.3). Many other verses of the *Rigveda* (I, 19.7; I, 23.17; I, 32.9) further explain the transfer of water
from earth to the atmosphere by the Sun and wind; breaking up of water into small particles and evaporation due
to Sun rays and subsequent rain; formation of cloud due to evaporation of water from the mother Earth and
returning in the form of rain. The verse I, 32.10 of the *Rigveda* further mentions that the water is never stationary
but it continuously gets evaporated and due to smallness of particles we cannot see the evaporated water particles.
According to *Atharvaveda* also (1200-1000 B.C.), the Sun rays are the main cause of rain and evaporation (Verse,
I, 5.2, in Sanskrit language):

amurya up surye yabhirg suryah sah| ta no hinvantvadhavaram||


The *Yajurveda* (1200 – 1000 BC) explains the process of water movement from clouds to Earth and its flow
through channels and storage into oceans and further evaporation (Verse, X, 19). During the time of *Atharvaveda*,
the concept of water evaporation, condensation, rainfall, river flow and storage and again repetition of cycle was



also well known as in the earlier Vedas. Therefore, it can be inferred that during the Vedic and earlier periods in
India, the concepts of infiltration, water movement, storage and evaporation as the part of hydrologic cycle were
well known to the contemporary Indian scholars.

The epic *Mahabharata* (Verse, XII,184.15-16) explains the water uptake process by plants and mentions that
rainfall occurs in four months (the Indian summer monsoon, ISM) (Verse, XII,362.4-5) and in the next eight
months (non-monsoon months), the same water is extracted by the Sun rays through the process of evaporation.
Likewise, in other Indian mythological scriptures such as *Puranas* (which are dated probably between 600 B.C.
to 700 A.D.), numerous references exist to hydrological cycle (NIH, 2018). The *Matsya Purana* (Verse, I, 54.29-
34) and *Vayu Purana* (Verse, 51.23-26) mention about the evaporation process which burns water by Sun rays
and is converted to smoke (i.e., process of evaporation) which ascend to atmosphere with the help of air and again
rains in next rainy season for the goodness of the living beings (NIH, 2018). The *Vayu Purana* and the *Matsya*
*Purana* also mention the rainfall potential of clouds and the formation of clouds by cyclonic, convectional and
orographic effects (Nair, 2004). Similarly, the *Linga Purana* (Verse, I, 36.67) clearly explains the various
processes of hydrologic cycle such as evaporation, condensation and mentions that water can't be destroyed; it
gets changed from one form to the other (NIH, 2018; Sharma and Shruthi, 2017) as:

jalasya nasho vridwirva natatyevasya vichartah| ghravenashrishthto vayuvrishti sanhrte punah||


The *Brahmanda Purana* (Verse, II, 9.138-139; 167-168) explains that Sun has rays of seven colors which extracts
water from all sources through heating (evaporation) and it gives to the formation of clouds of different colors
and shapes and finally these clouds rain with high intensity and great noise (NIH, 2018). The *Vayu Purana* also
refers to the various underground structures and topography such as lakes, barren tracts, dales, rocky rift valley
between mountains (Verse, 38.36).

The *Kishkindha sarga* (Chapter 28; Verses: 03, 07, 22, 27, 46) of the epic *Ramayana* discusses various aspects of
hydrological cycle. The verse 3 mentions about the formation of clouds by Sun and wind (through process of
evaporation from sea) and raining the elixir of life (water) and verse 46 mentions the overflowing of the rivers
due to heavy rains in rainy season. The verse 22 explains the process of cloud transportation laden with water and
elevational effects of the mountains on the whole processes. Based on these verses (and many more, not mentioned
here) a depiction on the various stages of the hydrologic cycle may also be established similar to Horton (1931).
Malik (2016) also compared the various concepts of modern hydrologic cycle with those presented in the
Ramayana and found that a corollary may be established between them.

The *Brihat Samhita* (literally meaning *big collection*) (550 A.D.) by Varahamihira, contains many scientific
discourses on the various aspects of meteorology, e.g., pregnancy of clouds, pregnancy of air,
winds, cloud formations, earthquakes, rainbows, dust storms and thunder bolts among other things such as colours
of the sky, shapes of clouds, the growth of vegetation, behaviour of animals, the nature of lightning and thunder
and associated rainfall patterns (Jha, 1988). The water falling from sky assumes various colours and tastes from
differences in the nature of Earth. Out of 33 chapters in the *Brihat Samhita*, 10 chapters are specifically devoted



to the meteorology. This highlights the depth of the meteorological knowledge prevalent during the period of
Varahmihira and his predecessors in the ancient India.

The verse 54.104 of *Brihat Samhita* explains the relation between soil and water. It is mentioned that pebbly and
sandy soil of copper color makes water astringent. Brown-colored soil gives rise to alkaline water, yellowish soil
makes water briny and in blue soil, underground water becomes pure and fresh. *Brihat Samhita* also discusses
about the geographical pointers such as plants, reptiles, insects as well as soil markers to gauge the groundwater
resources (occurrence and distribution) (Chapter 55, Dakargalam). It explains the groundwater recharge as "…
the water veins beneath the earth are like vein's in the human body, some higher and some lower..." as given in
the following verses (NIH, 2018):

Dharmyam yashashyam va vadabhaytoham dakargalam yen jaloplabdhiha

Punsam yathagdeshu shirastathaiva chhitavapi pronnatnimnasanstha.


Ekayna vardayna rasayna chambhyashchyutam namasto vasudha vishayshanta

Nana rastvam bahuvarnatam cha gatam pareekshyam chhititulyamayva.


The 'Dakargalam' (*Brihat Samhita*, Chapter 55) deals with ground water exploration and exploitation with various
surface features, that are used as bio indicators to locate sources of ground water, at depths varying from 2.29 m
to as much as 171.45 m (Prasad, 1980). The bio indicators, described in this ancient Sanskrit work, include various
plant species, their morphologic and physiographic features, termite mounds, geophysical characteristics, soils
and rocks (Prasad, 1986). All these indicators are nothing but the conspicuous responses to biological and
geological materials in a microenvironment, consequential to high relative humidity in a ground water ecosystem,
developed in an arid or semi-arid region. Variation in the height of water table with place, hot and cold springs,
groundwater utilization by means of wells, well construction methods and equipment are fully described in the
Dakargalam (Jain et al. 2007). It also means that the water which falls from the sky originally has the same colour
and same taste, but assumes different colour and taste after falling on the surface of Earth and after percolation.

Glucklich, (2008) opines about the *Brihat Samhita*: "… as the name of the work itself indicates, its data came
from numerous sources, some of them probably quite old. However, the prestige and systematic nature of the
*Brihat Samhita* gave its material the authority of prescriptions". Further, it is also appropriate to quote
Varahmihira (Chapter 1, Verse, II, Brihat Samhita) that '… having correctly examined the substance of the
voluminous works of the sages of the past, I attempt to write a clear treatise neither too long nor too short …'
(Iyer, 1884).

An interesting fact covered in details by Varahamihira is the role of termite knolls as indicator of underground
water. Apart from the underground water exploration, some of the verses of the chapter deal with topics such as
digging of wells, their alignment with reference to the prevailing winds, dealing with hard refractory stony strata,
sharpening and tempering of stone-breaking chisels and their heat treatment, treating with herbs of water with





objectionable taste, smell, protection of banks with timbering and stoning and planting with trees, and such other
related matters.

The Jain literature also made considerable contribution in the field of meteorology. The '*Prajnapana*' and
'*Avasyaka Curnis*' provide outstanding references to the various types of winds (Tripathi, 1969). The *Avasyaka*
*Curnis* furnishes a list of fifteen types of winds and the '*Prajanapana*' also mentions the snowfall and hailstorm
as form of the precipitation. The Buddhist literature also throws significant light on meteorology. In the narrative
of the first Jataka, named '*Apannaka*', several climatological facts are described therein. The Buddhist literature
refers to two general classes of clouds as: monsoon cloud and storm clouds or accidental ones (Tripathi, 1969).
The *Samyutta Nikaya* classifies clouds into five categories as (i) cool clouds, (ii) hot clouds, (iii) thunder clouds,
(iv) wind clouds –formed due to the activity of convection current in the atmosphere, and (v) rain clouds – most
probably cumulonimbus which brings copious downpour of rain.
**3 Measurement of Precipitation**
The "*Arthashastra*" and "*Astadhyayi*" of Panini (700 B.C.) mention about the rain gauges (Nair, 2004), which
was introduced by the Mauryan rulers in the *Magadha* country (south Bihar) in the fourth or third century B.C.
They are also credited with the establishment of first observatory. The system continued to be used by the
succeeding rulers until the end of the sixth century A.D. (Srinivasan, 1976). During the Mauryan period, the
raingauge was known as "*Varshamaan*". In the Arthshastra, the construction of the raingauge is described as "…
in front of the store house, a bowel (Kunda) with its mouth as wide as an aratni (24 *angulas* = 18" nearly) shall
be set up as raingague". However, the '*Arthshastra*' does not have any information about the height of the
raingauge (Srinivasan, 1976). This raingauge continued to be employed effectively by the succeeding rulers until
the end of the 600 A.D (Srinivasan, 1976; Murty, 1987). A schematic of the modern raingauge is shown in Figure
1. By comparing the dimensions of the ancient Indian and Symon's raingauge, one can infer about the advanced
level of knowledge possessed during that period.
The distribution of rainfall in various regions was well known during the Mauryan period. The '*Arthshastra*'
mentions as: "The quantity of rain that falls in the country of *jangiila* (desert regions or regions full of jungles) is
16 *dronas;* half as much more in *anupanam* (moist regions); as the regions which are fit for agriculture
*(desavapanam)*; 13.5 dronas in the regions of *asmakas* (Maharashtra); 23 *dronas* in Avanti (probably Malwa);
and an immense quantity in *aparantanam* (western regions, the area of Konkan); the borders of Himalayas and
the countries where water-channels are made use of in agriculture". Kautilya's method of classification of rainfall
areas in relation to the annual average quantity is indeed remarkable and he is the only classical author who treats
this aspect in a nutshell covering almost the whole of the Indian subcontinent (Srinivasan, 1976). From this, it is
evident that the methodology of measurement of rainfall given in *Arthshastra* is same as we have today, the only
difference is that rain was expressed in weight units. Discussing on the further geographical details of rainfall
variation, it is mentioned therein that "…when one third of the requisite quantity of the rainfalls, both during the
commencement and closing months of the rainy season, and two third in the middle, then the rainfall is considered
very even…".



The science of forecasting the rains had also come into existence as and must have been developing empirically.
It is further mentioned in the 'Arthshastra' that "the rainfall *forecasting can be made by observing the position,*
*motion and pregnancy (garbhadhan) of Jupitar, the rising, setting and motion of Venus, and the natural or*
*unnatural aspects of the Sun. From the movement of Venus, rainfall can be inferred".* Detailed descriptions on
classification of clouds and their water holding capacity (equivalent to the concept of atmospheric rivers) and
interrelationship of rainfall patterns and agriculture can also be found in the 'Arthshastra'.
Therefore, it can be concluded that during the Vedic era and afterwards in the age of epics and *Puranas*, (i.e.,
from 3000 B.C. to 500 A.D.), the knowledge of hydrologic cycle, ground water and water quality was highly
advanced, although the people of those times were solely dependent upon their experience of nature, without
sophisticated instruments of modern times. In the Vedic age, Indians had developed the concept that water gets
divided into minute particles due to the effect of Sun rays and wind, which ascends to the atmosphere by the
capillary of air and there, it gets condensed and subsequently falls as rainfall (*Vayu Purana*, 51. 14-15-16). The
*Linga Purana* also details on the various aspects of hydrological cycle (Sharma and Shruthi, 2017). Month wise
change in the facets of hydrological cycle was also known. Water uptake by plants which gets facilitated by the
conjunction of air along with the knowledge of infiltration is revealed in the ancient literature. In *Brihat Samhita*,
a separate chapter is devoted to the formation of clouds (*Garbhalakshanam*). A detail discussion has been given
on the properties of rainy seasons and their relationship with the movement of the planest and cloud formations
(Murthy, 1987). The *Brihat Samhita* also details on the measurement of rainfall and the dimensions of the
raingauge (Murty, 1987).
During the Mauryan period, it was possible to describe the distribution of rainfall in different areas of India.
Mauryans are credited with the installation of first observatory worldwide (Srinivasan, 1976). Modern
meteorological facts like arid region of Tibetan rain shadow area and no rainfall due to polar winds are fully
covered in *Puranas*. The Jain and Buddhist works guessed the actual height of clouds. Knowledge of monsoon
winds (Tripathi, 1969) and their effects as conceived by ancient Indians (*Brihat Samhita*) is in accordance to
modern hydro-science. These facts show that there was enriched knowledge of water science and associated
processes, including meteorology during ancient times in India, which is at par to the modern water science.
Well before many centuries of Christ, ancient Indians were aware of underground water bearing structures, change
in the direction of flow of ground water, high and low water tables at different places, hot and cold springs, ground
water utilization by means of wells, well construction methods and equipment, underground water quality and
even the artesian well schemes. This shows that well developed concepts of hydrological cycle, groundwater and
water quality were known to the ancient Indians in those ancient times while the contemporary world was relying
on the wild theories of origin and distribution of water.
**4 Water Management Technology in Ancient India**
The development of socio-cultural societies, agricultural establishments and permanent settlements led to the
establishment of a unique relationship between humans and water (Vuorinen et al., 2007; Lofrano and Brown,
2010). Scarborough (2003) and Ortloff (2009) discussed the impacts of water management practices on ancient
social structures and organizations with examples of the Eastern and Western hemispheres. Lofrano and Brown,



(2010) presented an in-depth review of wastewater management in the history of mankind. In this review work
they have categorically discussed about the evolution of sanitation through different civilizations of the world,
including the ancient Indus civilization).
During the Vedic age, the principle of collecting water from hilly areas of undulating surface and carrying it
through canals to distant areas was known (Bhattacharya, 2012). In the *Rigveda*, many verses indicate that the
agriculture can be progressed by use of water from wells, ponds (Verse, I, 23.18 and Verse, V, 32.2). Verse (VIII,
3.10) mentions construction of artificial canals by (Ribhus/Engineer) to irrigate desert areas. Verses (VIII, 49.6
and X, 64.9) emphasizes for efficient use of water, i.e., the water obtained from different sources such as wells,
rivers, rain and from any other sources on the earth should be used efficiently, as it is a gift of nature, for well-
being of all. There are also references of irrigation by wells (Verse, X. 25), canals (word '*kulya*' in *Rigveda*)
(Verse, X.99), and digging of the canal (Verse, X75) in the Rigveda. In *Mahâbhâsya* of Patañjali (150 B.C.) the
word '*kulya*' is also used.
Interestingly, the *Rigveda* (Verses, X 93.12; X 101.7) has a mention of 'asmacakra' (a wheel made of stones) and
water was raised with help of wheel in a pail using a leather strap. There is also a mention of '*Ghatayantra*' or
'*Udghatana*' (a drum-shaped wheel) round which a pair of endless ropes with ghata (i.e. earthen pots) tied at equal
distances. In Arabic literature, the water lifting wheel is also known as 'Noria'. Yannopoulos et al., (2015) also
mentioned that the ancient Indians had already developed water lifting and transportation devices. Further,
according to Joseph Needham (https://www.machinerylubrication.com/Read/1294/noria-history), due to evidence
documented in Indian texts dating from around 350 B.C., the 'Noria' was developed in India around the fifth or
fourth century B.C and transmitted to the west by the first century B.C. and to the China by the second century
A.D.
Similar to *Rigveda*, *Yajurveda* also contains references on water management. Verses VI, 100.2 and VII,11.1
mention "…that the learned men bring water to desert areas by means of well, pond, canals etc….and the man
should think about the drought, flood and like natural calamities in advance and take preventive measures
accordingly. Verse (XII,1.3) of *Atharvaveda* mentions that those who use rainwater by means of rivers, wells,
canals for navigation, recreation, agriculture etc., prosper all the time. Similarly, verse (XX, 77.8) of the
*Atharvaveda* directs the king to construct suitable canals across mountains to provide water for his 'subject' for
agriculture other purposes. The *Yajurveda* also has references, directing the man to use rain and river water by
means of wells, ponds, dams and distribute it to various places having need of water for agriculture and other
purposes. The *Atharvaveda* talks about the drought management through efficient use of available water resources
and emphasizes, these waters are used efficiently, will reduce the intensity of droughts. Verse (2.3.1) of the
*Atharvaveda* instructs for proper management of various water bodies such as brooks, wells, pools and an efficient
use of their waters resources for reducing the droughts intensity and water scarcity (Sharma and Shruthi, 2017).

As in many other parts of the World, civilization in India also flourished around rivers and deltas. Rivers remain
an enduring symbol of national culture (Nair, 2004). The Harappan (or Indus Valley) Civilization  (Figure 2)
which prospered during 2600–1900 B.C. (Chase et al., 2014) or about 5000 years ago (Dixit et al., 2018) had well



planned cities equipped with the public and private baths, well planned network of sewerage systems through
underground drains built with precisely laid bricks, and an efficient water management system with numerous
reservoirs and wells (Sharma and Shruthi, 2017). Evidences show that the Indus people developed one of the
smartest urban centres in those old times with exemplary fusion of civil, architectural and material sciences
(Possehl, 2002; Kenoyer, 1998; Wright, 2010). According to Shaw et al., (2007), the development of advanced
irrigation systems in ancient India led to the development of the complex urban societies and centres. The Indus
civilization was prominent in hydraulic engineering is known to have earliest known systems of flush toilets in
the world (Sharma and Shruthi, 2017). Kenoyer (2003) states that "… no other city in the ancient world had
developed such a sophisticated water and waste management system. Even during the Roman Empire, some 2,000
years later, these kinds of facilities were limited to upper-class neighbourhoods".
The Dholavira, an important city in the Indus civilization, contained sophisticated water management systems
comprising series of reservoirs, step wells and channels (Kirk, 1975; Sharma and Shruthi, 2017; Wright, 2010)
(Figures 3a and Figure 3b). The city is ringed with a series of 16 large reservoirs (7 m deep and 79 m long), some
of them interconnected and together, these storage structures account for about 10% of the area of the city (Iyer,
2019). The ability to conserve every drop of water in the parched landscape speaks volumes about the engineering
skills of the people of Dholavira. Recently, a rectangular stepwell has also been found at Dholavira which
measured 73.4 m long, 29.3 m wide, and 10 m deep, making it three times bigger than the Great Bath of Mohenjo-
Daro (https://www.secret-bases.co.uk/wiki/Dholavira).
The systems that Harappans of Dholavira city developed for conservation, harvesting, and storage of water, speak
eloquently about their advanced hydraulic engineering capabilities, given the state of technology (Baba et al.,
2018). The "Lothal" ("meaning *Mound of the dead*"), known as the harbour city of the Harappan civilization
(Bindra, 2003), is located at the *doab* of the Sabarmati and Bhogavo rivers. A roughly trapezoidal structure having
dimensions of 212.40 m on the western embankment, 209.30 m on the eastern one, 34.70 m on the southern one
and 36.70 m on the northern one (Rao, 1979) at Lothal is an example of advanced maritime activities in those old
days and is claimed by the archeologists to be the first known dockyard of the world (Nigam, 2016). Figure 4a
and Figure 4b show the dockyard at the Lothal after rains and the ancient Lothal as envisaged by the
Archaeological Survey of India (ASI). According to Nigam et al. (2016), the existence of the massive protective
wall (thickness up to 18 m) around the Dholavira city indicates the ancient Indians were aware of oceanic
calamities such as Tsunami/storm.

Agriculture was practised on a large scale having extensive networks of canals for irrigation (Nair, 2004). The
irrigation systems, different types of wells, water storage systems and low cost and sustainable water harvesting
techniques were developed throughout the region at that time (Nair, 2004; Wright, 2010). Mohenjo-Daro was one
of the major urban centres of the Harappan civilization receiving water from at least 700 wells and almost all
houses had one private well (Angelakis and Zheng, 2015). The wells were designed as circular to *pipal* (Ficus
religiosa) leaf shaped (Khan 2014). Canalising flood waters through ditches for irrigating the Rabi crops (crops
of the dry season) was also practiced at that time (Wright, 2010). The farmers of Harappa frequently used
"contouring, bunding, terracing, benching, *gabarbands* (dams) and canals for water management (Mckean, 1985).



The Gabarbands (stone-built dams for storing and controlling water) were also prevalent in these times for
irrigating agricultural lands during the dry seasons (Rabi crops) (Wright, 2010).

Agriculture and livestock rearing occupied a prominent role during Jainism and Buddhism period (600 B.C.) and
channel irrigation was in vogue (Bagchi and Bagchi, 1991). Field embankments were constructed surrounding the
fields to increase water holding capacity at strategic points with sluice gates to harness river water with proper
regulation facilities (*Arthshastra*, 400 B.C.) and irrigation through conduits was in practice to deliver water to the
irrigation field for attaining higher efficiency (Bagchi and Bagchi, 1991). Literature suggests that a large number
of hydraulic structures (dams, canals and lakes) were built during the Mauryan period in Indo-Gangetic plains and
other parts of the country for irrigation and drinking purposes (Shaw et al., 2007; Sutcliffe et al., 2011).
Surprisingly, many of these structures were equipped with the spillways to consider the flood protection measures.
During the Mauryan empire (400 BC-184 B.C), emperor Chandragupta Maurya constructed Sudarsana dam in
Girnar, Junagadh, Gujarat. Subsequent structural improvements involve the addition of conduits during the reign
of Asoka the Great, by his provincial governor the "Yavana Administrator (Greek Administrator)", Tusaspha
(Kielhorn, 1906; Shaw and Sutcliffe, 2001). In an excavation work conducted by Archaeological Survey of India
(ASI) during 1951-55, in Kumhrar (the site of ancient Pataliputra) a few miles south of Patna, Bihar "a canal 45
feet broad 10 feet deep and traced up to the length of 450 feet" was found of the Mauryan period. The canal was
linked with the 'Sone river' and also with the 'Ganges' for navigation purposes and also for the need of irrigation
to that area (Bhattacharya, 2012).
Here, it is instructive to quote Bhattacharya, (2012) : "… by the beginning of 300 B.C., a firm administrative set
up had taken shape. As a recognition of high position accorded to agriculture by the rulers as well as the people
at large, the construction of tanks and other types of reservoirs was considered to be an act of religious merit. The
king with the help and advice of his tiers of officials, ministers, consultants started acting as the "Chief trustee"
for optimizing, rationalizing and overall management of water resources. The *Arthasastra* of Kautilya gives us an
idea of principles and methods of management of irrigation systems … that the Mauryan kings took keen interest
in the irrigation schemes, is borne at by the report of Megasthenes (a Greek traveller) who mentions about a group
of officers responsible for superintending the rivers, measuring the land as is done in Egypt and inspecting the
sluices through which the water is released from the main canals into their branches so that everyone may have
an equal supply ...".
Shaw and Sutcliffe, (2001) presented hydrological background of the historical development of water resources
in South Asia with particular emphasis on ancient Indian irrigation system at the Sanchi site (a well-known
Buddhist site and a UNESCO World Heritage site located in Madhya Pradesh). They investigated a 16-reservoir
complex located in in the Betwa river sub-basin (a tributary of Yamuna in Ganga basin) in Madhya Pradesh, India
during 1998 and 2005 (Shaw, 2000; Shaw et al., 2007; Shaw and Sutcliffe, 2001, 2003a&b, 2005). In addition to
Sanchi, four other known Buddhist sites of Morel-khurd, Sonari, Satdhara and Andher, all established between
300-200 B.C. (Cunningham, 1854; Marshall, 1940) were also surveyed by them. The heights of the dams were
found to vary from 1 to 6 m and their lengths from 80 to 1400 m with flat downstream faces; presumably designed
to reduce damage from overtopping. At least two of the larger dams were equipped with spillways, which could



pass floods of about 50 years' return period and it suggests that flood protection was also taken into account while
designing these structures (Shaw and Sutcliffe, 2003a). Their reservoir volumes range from 0.03 to $4.7\times10^6$ m$^3$
and these estimates are closely related to the runoff generated by their catchments based on the present
hydrological conditions. These dams were constructed to a height sufficient to ensure that the reservoir volume
would be closely related to the volume of runoff from the upstream catchment of each site (Shaw and Sutcliffe,
2001). This indicates that these structures would have been constructed based on the detailed hydrological
investigations of the region. More or less identical spillways were also found with a group of much smaller
reservoirs in the neighbouring Devnimori area of Gujarat (Mehta, 1963). There are close similarities between the
Sanchi dams and well known Sudarsana dam (Shaw and Sutcliffe, 2003b). Sutcliffe et al., (2011) opines that it is
likely that some of the larger dams in the Sanchi area may have been fitted with similar spillways, which have
subsequently been obscured by siltation or erosion.
According to Shaw and Sutcliffe, (2001), a close relationship between runoff and reservoir volume in the Sanchi
area suggests a high level of understanding of water balance based on considerable period of observation and
understandings of local conditions. While excavating the area around the 'Heliodorus' pillar in Vedisa (present
day Vidisha, Madhya Pradesh), Bhandarkar, (1914) found the remains of a 300 B.C. canal, which would have
been drawing water from the river Betwa. However, Shaw and Sutcliffe, (2001) further mentions that a more
comprehensive understanding of ancient Indian irrigation would have been developed; had adequate attention
been paid to the Sanchi reservoir complex during the Vedisa excavations. Based on these findings, Shaw and
Sutcliffe (2003a&b) and Sutcliffe et al. (2011) conclude that the Sanchi Dam system would have been built on
the basis of a sound knowledge of the principles of water balance with detailed hydrological investigations and
by 'engineers with experience of reservoir irrigation' with a higher level understanding of the hydraulic
technology.

During the Sangam Period (300 B.C. to 300 A.D.), in the southern parts of India, the rainwater harvesting
structures such as tanks (*ery* in Tamil) were constructed for irrigating the paddy fields (Fardin et al., 2013; Sita,
2000) and fishing was also practiced in lotus ponds (*tamaraikulam* in Tamil) (Sita, 2000). The Grand Anicut
(Kallanai Dam) was constructed by the Chola King Karikalan during the 1$^{st}$ century A.D. on the river Cauvery for
protection of the downstream populations against flood and to provide for irrigation supplies in the Cauvery delta
region. The Grand Anicut is the world's oldest still in use dam and is also credited with being the 4$^{th}$ oldest dam
in the world and the first in India. In *Brihat Samhita* (550 A.D.), there are references regarding the orientation of
ponds, bank protection through pitching, plantation and also by providing sluicing arrangements. *Brihat Samhita*
contains many references regarding the orientation of ponds so as to store and conserve water efficiently (reducing
evaporation losses), plantation type for bank protection and proper sluicing to protect pond/reservoir from any
possible damage. Verse (54.118) mentions that a pond oriented in east to west direction retains water for a long
time while one from north to south loses invariably by the waves raised by the winds. Verse (54.120) suggests for
construction of spillway as an outlet for the water should be made on a side with the passage being laid with
stones.



## 5 Wastewater Management in Ancient India

The sanitation and wastewater management has always been one of the most important socio-environmental challenges that the humankind has ever faced and the societies in the ancient India had developed stat-of-the art technological solutions by utilizing their knowledge on hydraulic systems with the structural and materials advancements. Apart from the detailed references on various aspects of hydrology as discussed earlier, we also get some references to water quality in Vedas and other early literature, especially in *Atharvaveda, Charaka Samhita*, and *Susruta Samhita* (both of pre- or early Buddhist era) (NIH, 2018). There are hymns in *Rigveda* stating the role of forest conservation and tree plantation on water quality (Verse V, 83.4). The Verse V, 22.5 of *Atharvaveda*, cautioned people from diseases living in a region with heavy rainfall and bad quality of water. There are instances of classifying water based on taste in epic *Mahabharata* (Verse XII, 184.31 & 224.42). The *Brihat Samhita* also discussed the relationship between soil colour and water quality (Verse, 54.104) and techniques are mentioned for obtaining potable water with medicinal properties from contaminated water (Verses 54.121 & 54.122).

The Harappan cities were one of the very first and most urbanised centres developed with the excellent civil and architectural knowledge in the old world. Even as early as 2500 BCE, Harappa and Mohenjo-Daro included the world's first urban sanitation systems (Webster, 1962). The water and wastewater management systems have been highly amenable to the socio-cultural and socio-economic conditions and religious ways of societies through all the ages of the civilizations (Sorcinelli, 1998; Wolfe, 1999; De Feo and Napoli, 2007; Lofrano and Brown, 2010). All through the ages, the wastewater management has been considered filthy (Lofrano and Brown, 2010; Maneglier, 1994), The evolution process of wastewater management through the ages has been discussed by several researchers worldwide, (e.g., Maneglier, 1994; Serneri, 2007; Sorcinelli, 1998; Sori, 2001; Tarr, 1985; Viale, 2000). Recently, Lofrano and Brown, (2010) presented an in-depth review of wastewater management in the history of mankind and found that the 'Indus civilization was the first to have proper wastewater treatment systems' in those ancient times. Wastewater management and sanitation were the major characteristics of the first urban sites of the Harappan civilisation (Kenoyer, 1991). The sewage and drainage systems were composed of complex networks, especially in Mohenjo-Daro and Harappa (Jansen, 1989). Latrines, soak-pits, cesspools, pipes and channels were the main elements of wastewater disposal (Fardin et al., 2013).

All the houses were connected to the drainage channels covered with bricks and cut stones and the household wastewater was first collected through tapered terra-cotta pipes into the small sumps for sedimentation and removal of larger contaminants (primary wastewater treatment) and then into drainage channels in the street. This most likely was the first attempt at treatment on record (Lofrano and Brown, 2010). These drainage channels were having the provision of cleaning and maintenance by removing the bricks and cut stones (Wolfe, 1999). The cesspits were fitted at the junction of the several drains to avoid the clogging of the drainage systems (Wright, 2010). Fardin et al., (2013) mention that almost all the settlements of Mohenjo-Daro were connected to the drain network. However, at the same time, at Kalibangan, toilets and bathrooms outflows were connected in U-shaped channels made of wood or terracotta bricks with decentralised sewage systems. These effluents poured into a jar placed in the main street (Chakrabarti, 1995). The same model of wastewater collection was used in Banawali, where effluents were channelled into drains made of clay bricks, before reaching the jars (Bisht, 1984).





In many other parts of the ancient India, e.g., Jorwe (Maharashtra), a similar drainage system was established
during 1375–1050 BC (Fardin et al., 2013; Kirk, 1975); at around 500 B.C., the city of Ujjain was also laid down
with the sophisticated drainage system having soak-pits built of pottery-ring or pierced pots (Kirk, 1975; Mate,
1969) and in Taxila around 300 B.C. very muh similar drainage system to that of Mohenjo-Daro was in place.
(Singh, 2009). This shows that during the ancient times, modern concepts of sanitation and waste water
management technology were very well known to the Indians and were in their advanced stages during the Indus
valley civilization and later periods. Modern methods of wastewater disposal systems based on centralized and
decentralized concept as well as methods for wastewater treatments during Indus valley civilization were even
better than those used in the contemporary world.
**6 Summary and Conclusions**
This paper has explored the hydrological developments in ancient India starting from Harappa Civilization to the
Vedic Era and later, using references from Vedas, mythological epics such as *Mahabharata*, *Ramayana*, Jain and
Buddhist literature, and the references of *Arthshastra*, Astadhyayi and many other Vedic text such as Puranas
(*Brahmana*, *Linga*, etc.), *Brihat Samhita*, and other ancient literature. The following conclusions can be drawn
from this exploration:
1. The Vedas, particularly the *Rigveda*, *Atharvaveda* and *Yajurveda* had specifically dwelt upon the hydrologic
cycle and various associated processes. The concepts of evaporation, cloud formation, water movement,
infiltration and river flow and repetition of cycle are explicitly discussed in these ancient texts. *Ramayana*
has also mentioned about hydrologic cycle and artesian wells. *Mahabharata* explains about the monsoon
seasons and water uptake process by plants. Rigveda also mentions about water lifting device such as
*Asmacakra/Ghatyanta* (similar to Noria), among others.
2. *Matsya Purana, Vayu Purana*, *Linga Purana*, and *Brahmanda Purana* also mention about the processes of
evaporation, formation of clouds due to cyclonic, convectional and orographic effects, rainfall potential of
clouds and many other associated hydrological processes.
3. The *Rigveda, Atharvveda, Brihat Samhita, Susrutu Samhita* and *Charaka Samhita* have numerous references
of water quality and nature-based solutions (NBS) for obtaining potable water. The Dakargalam Chapter of
*Brihat Samhita* dwelt upon the occurrence and distribution of groundwater resources using geographical
pointers and soil markers.
4. The Harappa Civilization epitomizes the level of development in water sciences. Extensive network of canals,
water storage structures, different types of wells, and low cost and sustainable water harvesting structures
were developed during this period. These people had created sophisticated water and wastewater management
systems, planned network of sewerage systems through underground drains and also had the earliest known
system of flush toilets in the world. The Harappa Civilization is also credited with the first known dockyard
in the entire world. Indus people were also aware about the oceanic calamities such as Tsunami.



5. The first observatory for measuring rainfall using '*Varshamaan*' (raingauge) was established during Mauryan empire in India. The reservoirs, dams, canals equipped with the spillways were constructed for irrigation and domestic supplies with adequate knowledge of water balance. Some structures were also constructed considering 50 years' return period. In ancient water history, the Mauryan period is also credited with the first and foremost hydraulic civilization. Forecasting of rainfall and water pricing system was also prevalent in this period.

6. The hydrologic knowledge in ancient India was contained in the *shlokas* of scriptures and very few people are conversant with the languages of the scriptures. Hence, the knowledge and wisdom remained largely unknown to the recent generations. Further, the script of the Harappans has not yet been deciphered. If further research is carried out on ancient literature and when the script of the Harappans is deciphered, it is highly likely that many more facts will emerge which may be much more fascinating than what we know so far.

**Data availability.** No data sets were used in this article.

**Author contributions.** PKS, PD and SKJ developed the structure of the paper. PKS wrote the paper and PD contributed to Sect. 5. PD also contributed to referencing and formatting the manuscript. SKJ and PPM reviewed and supervised the content of the manuscript.

**Competing interests.** The authors declare that they have no conflict of interest.

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

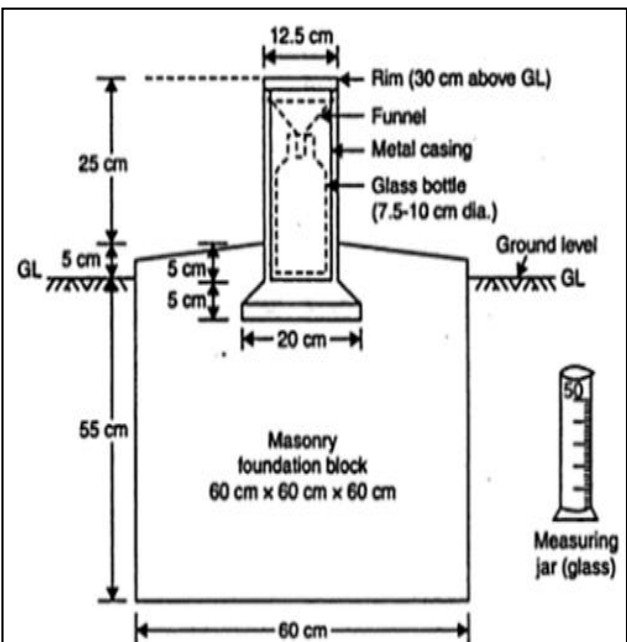


**Figure 1: The Symon's raingauge [Source: Raghunath, (2006)].**




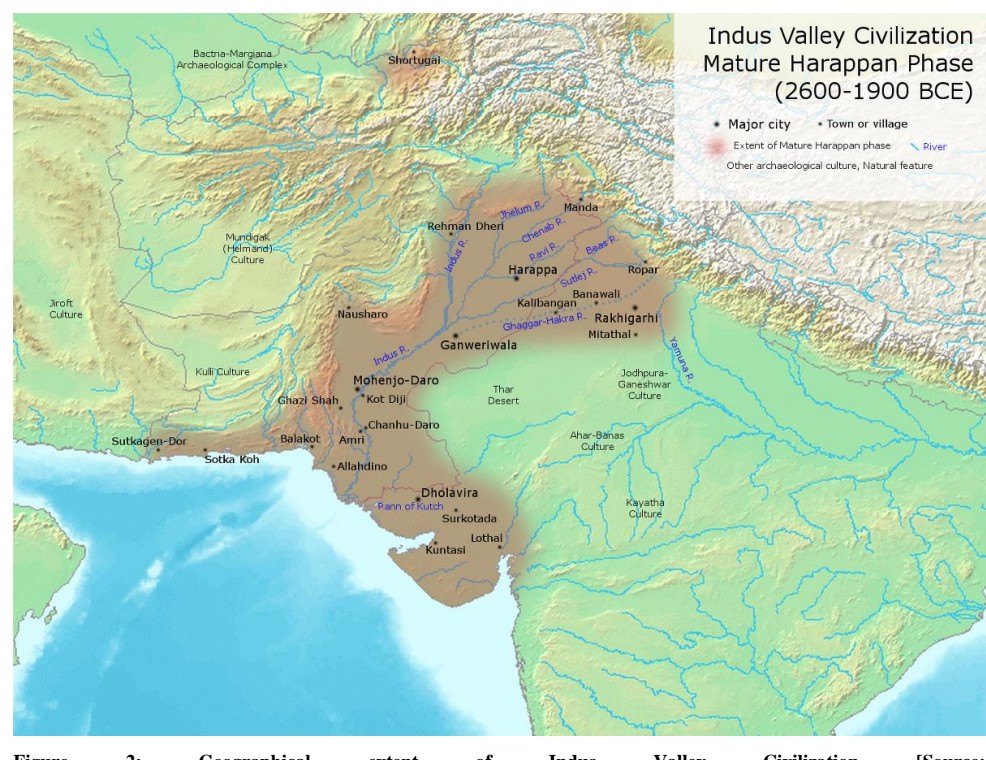

**Figure 2: Geographical extent of Indus Valley Civilization [Source: https://commons.wikimedia.org/wiki/File:Indus_Valley_Civilization,_Mature_Phase_(2600-1900_BCE).png].**

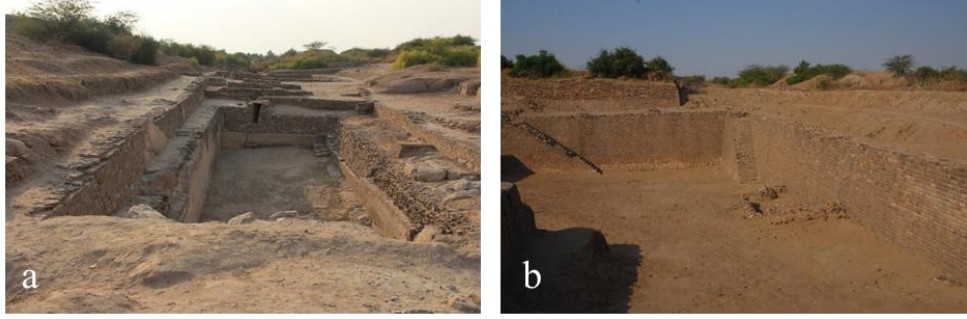

**Figure 3: The southern (a) and eastern (b) reservoirs of Dholavira [Source: Iyer, (2019)].**



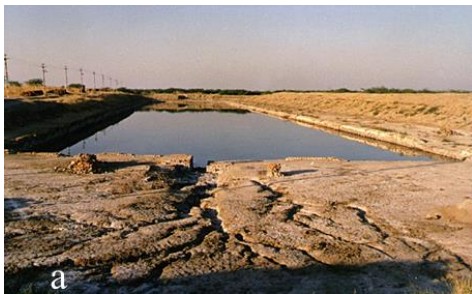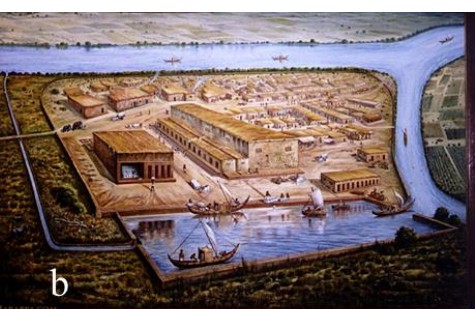

**Figure 4: Dockyard (a) and ancient Indus port (b) of Lothal [Source: https://www.harappa.com].**