# Peer review of "Hydrology and Water Resources Management in Ancient"

_Hydrology and Earth System Sciences, 2020_

## Referee Comment (RC1) · Stefano Barontini (Referee) · 8 Jun 2020

I read with interest the contribution *Hydrology and water resources management in ancient India* by Singh et al., in which, on the basis of an accurate bibliographical review, the Authors present many aspects of the multifaceted hydrological and hydraulic knowledge in ancient India. The themes addressed are the comprehension of the hydrological cycle, the precipitation measurements, the water management (with more evidence to the hydraulic structures than to the management practices) and the wastewater management.

The paper is well written and thoroughly argumented, and it makes a state of the art of the matter, provided that the topic stands between many disciplines (history, aschaeology, hydraulic engineering, history of technology and history of culture). Therefore the paper might be eventually recommended for publication, but I encourage the Authors to strengthen its unitary perspective, in order to depict a wide portrayal, thus avoiding the risk of giving the idea of a collection of cases.

As a first point, for example, it might be useful to explicitly state both in the Abstract and in the Introduction which are the geographical and historical boundaries of the matters, and possibly why these boundaries were chosen, and the aims and the methods of the research.

Many informations presented in the Introduction might be effectively contextualized in the following sections, whereas in the Introduction it is recommended to declare which is the order along which the matter is presented in each section (e.g. historical order, or process– or technology–based order, etc.).

Also the concept of "hydraulic civilization", which is sometimes used in the paper, might be better defined in the Introduction. In fact in all the ancient and modern societies the water management plays a crucial role, but the attribute of "hydraulic civilization" is nowadays preferably used to identify those civilizations which survival was deeply linked with the capability of managing the water–related issues (as e.g. the water scarcity, the soil salinization, or the floods) and, in most of the cases, the management was centralized via well structured groups of technicians and skilled workers (as it was e.g. the case of the great Central Asia oases).

Finally I encourage the Authors to enlighten, on the basis of the investigated literature, the links between the Indian hydraulic culture and that of the surrounding cultures, particularly regarding the water technologies (see below for details).

As a general typographical aspect, I recommend to check and uniform all the emphases and the citations, and to add a complete English translation to all the book titles (the first time they are introduced) and to all the ancient citations.

In the following I list some detailed notes:

**line 49** add a reference for the citation;

**l.53** emphasize *variyantra* and better detail its functionning;

**l.57** *pynes* and *ahars* are very interesting structures, also in this case I recommend to better define their functionning (e.g. whether *ahars* are fed by *pynes* or by the slopes) and, if possible, their diffusion;

**l.73** it is meant the *Arthashastra* of l.50, isn't it?

**l.115** *it can be inferred...*: this is an important point for the comprehension of the hydrological cycle.

Since what it is reported, it seems that the correct comprehension of the hydrological cycle was already achieved in ancient India, as it was few centuries later in ancient Greece, before the Aristotelian statement according to which the water of great rivers could not be stored inside the Earth. Are there explicit references to issues related to the infiltration and to the storage in subsoil reservoirs?

This conjecture (the Aristotelian one) paved the way to an (uncorrect) description of the hydrological cycle based on the concurrence of two cycles: one external to the Earth, driven by the Sun, and a more important one internal to the Earth, driven by an engine placed within the Earth's depths. At Authors' knowledge, are there reflections of this conjecture in the Indian late–antiquity hydrological culture?

Moreover *Puranas* are reported to be written between 600 B.C. and 700 A.D.: is it possible to provide a closer time range for the ones which are cited by the Authors (and particularly for the *Vayu Purana*)?

[Figure]

**l.125** Do ancient texts use the word *smoke* instead of *vapour*? It might be interesting, as in the Aristotelian tradition *smoke* is used for the dry air in opposition to *vapour* which is used for the moist one;

**l.132** Add an English translation (as well for the other citations and titles, see before in the general comment);

**ll.162—163** It is a very interesting point, as the *veins* metaphore was common also in other contexts (see e.g. Leonard from Vinci). What feds such veins, as it is reported by *Brihat Samhita*? And which is the direction along which do they flow?

**ll.216—217** Probably not necessary;

**l.223** Kautilya. . . : add a reference;

**l.231** It seems an astronomical approach, rather than an empiristic one: were there found evidences for multiannual precipitation cycles?

**l.242** Please, check whether *capillary* is properly used;

**l.257** In which sense it is used *change in the direction of flow of groundwater*?

**l.260** Artesian wells seems not been introduced before, a reference will be useful;

**l.267** In which sense are introduced Eastern and Western emispheres?

**ll.281—282** It seems more a *saqiya* than a *naoor / noria*: could the Authors add few details?

**l.285 and followings** Probably it is not necessary to enter here the debate on the origin of the noria, or it is better to strengthen the cited references base on this topic;

**l.336** In which sense *low cost* is used?

**l.340 and followings** *Rabi* irrigation was a spate irrigation, a basin irrigation, or a furrows irrigation?

**l.364** *. . . an act of religious merit*: it is very interesting to unveil the cultural link between the humans and the Nature. Could the Authors better detail in which sense building reservoirs was considered a religious merit?

**l.379** These dams seems more *barrages*, eventually used also for spate irrigation. Could the Authors add some more details on the discharge regime and on the use of these dams? Is it a *wadi* regime?

**l.381** Is the return period referred to present climate or it was estimated for the ancient one?

**ll.434—440** Probably not necessary here, and more useful in the Introduction;

**l.447** *tapered terra–cotta pipes*: Could the Authors add some details on these pipes? They seem frustum–of–cone shaped *fistulae* common in the Central Asia oases and Latin world;

**ll.463—465** It sounds not very clear, probably not necessary.

---

## Author Comment (AC1) · 8 Jul 2020

We thank the reviewer, Prof. Stefano Barontini, for offering valuable suggestions and comments to improve the manuscript. We have greatly benefited by by the comments. We provide here our responses to the comments and mention the actions taken where relevant.

Response to the Reviewer's Comments

General Comment : I read with interest the contribution Hydrology and water resources management in ancient India by Singh et al., in which, on the basis of an accurate bibliographical review, the Authors present many aspects of the multifaceted hydrological and hydraulic knowledge in ancient India. The themes addressed are the comprehen-

sion of the hydrological cycle, the precipitation measurements, the water management (with more evidence to the hydraulic structures than to the management practices) and the wastewater management.

The paper is well written and thoroughly argumented, and it makes a state of the art of the matter, provided that the topic stands between many disciplines (history, archaeology, hydraulic engineering, history of technology and history of culture). Therefore, the paper might be eventually recommended for publication, but I encourage the Authors to strengthen its unitary perspective, in order to depict a wide portrayal, thus avoiding the risk of giving the idea of a collection of cases.

Response: We thank the reviewer for the positive feedback and for offering several comments to improve the manuscript. We have greatly benefited by the comments of the reviewer. We provide here our responses and mention how we would modify the manuscript.

Comment 1: As a first point, for example, it might be useful to explicitly state both in the Abstract and in the Introduction which are the geographical and historical boundaries of the matters, and possibly why these boundaries were chosen, and the aims and the methods of the research.

Response 1: The geographical region covers the entire Indian sub-continent to the east of the Indus river. It includes the parts of the Harappan civilization (in the present-day Pakistan) and entire India. These boundaries encompass the major centers/regions of the development in the ancient India. It would be appropriate to quote Olson (2009) here: 'India was not re-united for nearly 500 years after the collapse of the Mauryan Empire, so its end forms a logical place to end the discussion of the ancient India'. Our discussion in the manuscript is mainly concerned with this period.

This discussion will be incorporated in the revised version of the manuscript, and the geographic region will be mentioned in the Abstract, as suggested by the reviewer.

Comment 2: Much information presented in the Introduction might be effectively con-textualized in the following sections, whereas in the Introduction it is recommended to declare which is the order along which the matter is presented in each section (e.g. historical order, or process– or technology–based order, etc.).

Response 2: The manuscript has been prepared in view of the process-or technology-based order. While doing so, the historical order of those processes or technologies has also been maintained in the manuscript. The text of the manuscript in each section (including the Introduction) will be re-structured accordingly.

Comment 3: Also the concept of "hydraulic civilization", which is sometimes used in the paper, might be better defined in the Introduction. In fact in all the ancient and mod-ern societies the water management plays a crucial role, but the attribute of "hydraulic civilization" is nowadays preferably used to identify those civilizations which survival was deeply linked with the capability of managing the water–related issues (as e.g. the water scarcity, the soil salinization, or the floods) and, in most of the cases, the man-agement was centralized via well-structured groups of technicians and skilled workers (as it was e.g. the case of the great Central Asia oases).

Response 3: Thank you for this insightful comment. In this manuscript, the concept of hydraulic civilization was referred to in respect of the Harrapa civilization and the Mauryan Empire (321-297 BC) in India. Following discussion will be added in the revision, with appropriate editing.

"According to McClellan III and Dorn (2006), the Mauryan Empire was 'first and fore-most a great hydraulic civilization.' Megasthenes (A Greek traveller in Chandragupta's Court, around 300 BC), mentions that 'more than half of the arable land was irrigated and in agriculture and produced two harvests in a year'. Further, there was a special department for supervision, construction and maintenance of a well-developed irriga-tion system with extensive canals and sluices, wells, lakes and tanks. The same bureau was responsible for planning and settlement of the uncultivated land. A similar description of the different institutional arrangements during Mauryan period can be had from Arthasastra. The importance of the hydraulic structures in the Mauryan period can be adjudged on the basis of the punishments/fines to the offenders. As mentioned in the Arthasastra, 'When a person breaks the dam of a tank full of water, he shall be drowned in the very tank; of a tank without water, he shall be punished with the highest amercement; and of a tank which is in ruins owing to neglect, he shall be punished with the middle-most amercement'.

Remarkably, the Mauryan Empire did not lack the other hallmarks associated with the hydraulic civilizations (McClellan III and Dorn, 2006). It had the departments concerned with the rivers, excavating and irrigation along with a number of regional and other superintendents such as the superintendent of rivers, agriculture, weights and measures, store-house, space and time, ferries, boats, and ships, towns, pasture grounds, road-cess, and many others along with many strata of the associated officers such as head of the departments (adhyakshah), collector-general (samahartri), and chamberlain (sannidhatri), etc. Olson (2009) also mentions that there was an extensive irrigation network organised by a state bureaucracy. According to Wittfogel (1955), the Mauryan Empire had virtually all of those characteristics that a hydraulic civilization must possess (though it was late and short lived).

Water pricing was very well defined in the Mauryan Empire. According to Arthasastra, those who cultivate irrigating by manual labour (hastaprávartimam) shall pay 1/5th of the produce as water-rate (udakabhágam); by carrying water on shoulders (skandhaprávartimam) 1/4th of the produce; by water-lifts (srotoyantraprávartimam), 1/3rd of the produce; and by raising water from rivers, lakes, tanks, and wells (nadisarastatákakúpodghátam),1/3rd or 1/4th of the produce.The Superintendent of the Agriculture was responsible for compiling the meteorological statistics by using a rain gauge and for observing the sowing of the wet crops, winter crops or summer crops depending on the availability of the water."

Comment 4: Finally, I encourage the Authors to enlighten, on the basis of the investigated literature, the links between the Indian hydraulic culture and that of the surrounding cultures, particularly regarding the water technologies (see below for details).

Response 4: A separate section, with appropriate editing, will be added in the revised manuscript.

Hydraulic Inter-linkages between the Ancient Indian and Nearby Cultures

All the ancient civilizations, i.e., Harappan, Egyptian, Mesopotamian, Chinese, and including the Minoan civilization that flourished and attained their pinnacle were largely dependent on degree/extent of their advancements in the field of water technologies. With the efficient management of water resources, they were able to produce more food grains and mitigate the damages due to natural hazards such as droughts and floods. At the same time, the advanced wastewater management techniques helped in healthy lifestyles, hygiene, and clean environments.

The ancient Indian literature, starting from the Harappan civilization to the Vedic Period followed by the Mauryan Empire, the Vedic Samhitas and Puranas, contains detailed discourses on the various processes of hydrological cycle, including groundwater exploration, water quality, well construction, irrigation by channels (kulya). Water technological advancements coupled with the architectural sophistication during the Harappan civilization were at their zenith. Nowhere in the world we had such sophisticated and impressive planning relating to the water supply and effluent disposal system (Jansen, 1989). Almost all houses were having their private wells with bath and toilet area lined with the standard size burnt bricks and draining into the soak pit or into the street drains.

Multiple flushing lavatories attached to a sophisticated sewage system were located in the ancient cities of Harappa and Mohenjo-Daro civilization (Pruthi, 2004). The Great Bath at Mohenjo-Daro and 16 reservoir system of the Dholavira and the Dock yard are the perfect examples of the excellent hydraulic engineering in the Harappan civilization. The Mauryan Empire was named as the 'hydraulic civilization' due to developments of

the advanced means of irrigation, construction of wells, dams and reservoirs, rainfall measurements, protection of hydraulic structures, and water pricing systems in place and a stratified establishment of the bureaucratic and engineering establishment.

The effluent disposal drainage systems were well-known to almost all the civilizations at that time with varying level of technological advancements. The Egyptian civilization (∼2000-500 BC), lacked the flushing lavatories and sophisticated sewer and wastewater disposal systems at that time as was prevalent in Harappan. The copper pipes were in use in some Pyramids for building bathrooms and sewerage system (De Feo et al., 2014). The Mesopotamian civilization (ca. 4000–2500 BC) also had well-constructed storm drainage and sanitary sewer systems. However, there seems no system of vertical water supply by means of wells and it was even practically unknown in the early urban cultures (Jansen, 1989; De Feo, 2014). According to Jansen (1989) and De Feo et al., (2014), the very efficient drainage and sewerage systems, flushing toilets, which can be compared to the modern ones, re-established in Europe and North America in a century and half ago.

The Mohenjo-Daro city was serviced by at least 700 wells, whereas, the contemporary Egyptian and Mesopotamians had to fetch water bucket-by-bucket from the river and then store in the tanks at homes (Jansen, 1989). The bathing platforms in the Harappan civilizations were also unique as compared to the Mesopotamian and other civilizations. The ancient cities of the Mesopotamian civilization, i.e., UR and Babylone had effective drainage system for storm water control, sewers and drains for household waste and drains specifically for surface runoff (Jones, 1967; Maner, 1966). The ancient Mesopotamians had also developed canal irrigated agriculture and constructed dams across the Tigris river for diverting water to meet the irrigation and domestic supplies. The 'qanat' were widely used in Mesopotamian civilization for transferring the water from one place to another using the gravity. The urban centers of the Sumer (Sumerian) and Akkud (Akkadian) (third millennium BC) had water supplies by canal(s) connected to the Euphrates River. However, this lacks the advancements as compared

to the Harappan civilization. The water lifting device were also used in Mesopotamian Civilization and the Saaqia (or water wheel) was widely used for lift irrigation using oxen for irrigating the summer crops (Mays, 2008).The 'asmacakra' and 'Ghatayantra' were widely in use during the Vedic and Mauryan Period. The 'Varshaman' was widely used in Mauryan Empire for rainfall measurements. It may be noted that we do not have any reference of 'rainfall measurement' in other contemporary civilizations in the old world. The Pynes-Ahar system of participatory irrigation and rainwater harvesting is a unique system developed in Ancient India.

In Chinese (Hwang-Ho) civilization, the Shang dynasty (1520-1030 BC) developed extensive irrigation works for rice cultivation. Various water works such as dikes, dams, canals and artificial lakes proliferated across the Chinese civilization. Yu the Great, is acclaimed in China as the 'controller of the waters'. During the period 1100-221 BC, the Lingzi city (covering an area of 15 km2) also had a complex water supply and drainage system, combined with the river, drainage raceway, pipeline and moat (De Feo, et al., 2014). The moat surrounding the town halls had supplies from the river works as daily water uses. The water-fortification (audaka) around the forts was also a prime requirement in the Mauryan Empire. Notably, the drainage system of the Lingzi town is supposed to be the oldest and biggest in the ancient China (Fan, 1987). The drainage systems to collect rainwater and wastewater into pools and finally discharge into river were made of the earthenware pipes. The underground urban drainage systems were also in existence in Chine during the Shan Dynasty (∼10-15 BC).

The Minoan civilization (∼3200-1100 BC) is considered to be the first and the most important European culture (Khan et al., 2020). The Crete island was the center of the Minoan civilization and was known for architectural and hydraulic operation of its water supply, sewerage, and drainage systems (Khan et al., 2020). Aqueducts made of terracotta were in use for transporting water from the mountain springs. Water cistern were used for storing rainwater and spring water for further transporting it by using aqueduct. Lavatories with the flushing system were also in use in this civilization. In words

of Jansen (1989), 'for the first time in the history of mankind, the waterworks developed in Harappan civilization were to such a perfection which was to remain unsurpassed until the coming of the Romans and the flowering of civil engineering and architecture in classical antiquity, more than 2,000 years later'.

Overall, if we closely look at the scale of the hydro-technologies in all the civilizations, the Harappan civilization is not only credited with the more advanced and larger scale application of hydro-technologies (hydrologic, hydraulic and hydro-mechanical) but also worked as a 'landmark' for the contemporary civilizations to achieve the great heights in human civilizations, on the whole.

Comment 5: As a general typographical aspect, I recommend to check and uniform all the emphases and the citations, and to add a complete English translation to all the book titles (the first time they are introduced) and to all the ancient citations.

Response 5: Yes, the suggestions will be taken care of in the revised version of the manuscript.

Comment 6: line 49 add a reference for the citation;

Response 6: The reference, Mujumdar and Jain (2018) will be added.

Comment 7: l.53 emphasize variyantra and better detail its functioning;

Response 7: A revised sentence is given here, which will be added in the manuscript: The variyantra (water machine) was similar to the water cooler. According to Megasthenes (an ancient Greek historian in the court of King Chandragupta Maurya), the variyanytra was used by the wealthier sections of the society for cooling the air.

Comment 8: l.57 pynes and ahars are very interesting structures, also in this case I recommend to better define their functioning (e.g. whether ahars are fed by pynes or by the slopes) and, if possible, their diffusion;

Response 8: Thank you for the suggestion. The Pynes are man-made channels to

utilize the river water flowing through the hilly rivers of South Bihar and Chhota Nagpur plateau, whereas the Ahars are catchments with embankments on three sides to store rainwater and the water from the Pynes (Naz and Subramanian, 2010). The Ahar-Pyne system is still widely practiced in these regions and it is a shining example of participatory irrigation management (Pant and Verma, 2010). The Pynes feed many Ahars and several distributaries are then constructed from both Pynes and Ahars for irrigating the field (Sengupta, 1985; Verma, 1993). The Ahar-Pyne system is extremely suitable for the regions having scanty rainfall, highly undulating and rocky terrain, soils with heavy clay or loose sand (lower moisture holding capacity) and steep slope thus causing extensive surface runoff.

The Pynes are of different sizes. If the Pynes are originating from the Ahars, then these are smaller in size (3 to 5 km) and used for irrigating cultivable fields, where as if these originating from the rivers, then the size may vary from 16 to 32 km in length and some of them known as dasianpynes (pynes with 10 branches) to irrigate many thousand acres of the land (O' Malley, 1919). Apart from participatory irrigation system, the Ahar-Pyne system also works as flood mitigation system (Roy Choudhry, 1957). Worth mentioning, recently the Government of Bihar has started the 'renovation' of the traditional water bodies (Ahar-Pyne system) under 'Jal Jeevan Hariyali' programme. This reflects the importance of this ancient hydraulic structure for water harvesting even in the modern times in India (as shown in Figure 1).

We will add a brief discussion on this in the manuscript.

Comment 9: l.73 it is meant the Arthashastra of l.50, isn't it?

Response 9: Thank you. Yes, it is same as in line # 50, i.e., (Arthashastra). The meaning of the Arthashastra is the 'the science of material gain'.

Comment 10: l.115 it can be inferred. . . : this is an important point for the comprehension of the hydrological cycle.

a. Since what it is reported, it seems that the correct comprehension of the hydrological cycle was already achieved in ancient India, as it was few centuries later in ancient Greece, before the Aristotelian statement according to which the water of great rivers could not be stored inside the Earth. Are there explicit references to issues related to the infiltration and to the storage in subsoil reservoirs?

b. This conjecture (the Aristotelian one) paved the way to an (uncorrect) description of the hydrological cycle based on the concurrence of two cycles: one external to the Earth, driven by the Sun, and a more important one internal to the Earth, driven by an engine placed within the Earth's depths. At Authors' knowledge, are there reflections of this conjecture in the Indian late–antiquity hydrological culture?

c. Moreover, Puranas are reported to be written between 600 B.C. and 700 A.D.: is it possible to provide a closer time range for the ones which are cited by the Authors (and particularly for the Vayu Purana)?

Response 10:

a. The infiltration process and sub-soil reservoirs is defined in the Brihat Samhita (550 AD) as given in Line # 162-163. However, the Verses 184.15-17 of Mahabharata state that the plants drink water through their roots. It is said that the water uptake process is facilitated by the conjunction of air. b. The 'Sun' is the main source of the hydrologic cycle [Lines # 107-108; Page# 3] was very well know from the days of Vedic periods. In Rigveda [Lines 100-101; Page #3 of the manuscript], it is mentioned therein that 'the God has created ''Sun' and has placed it in such a position........". c. The Puranas are a class of literary texts, all written in Sanskrit verse, whose composition dates from the 4th century BCE to about 1,000 A.D (http://southasia.ucla.edu/religions/texts/puranas/). Further it would be interesting to quote Dimmitt and van Buitenen (1978): "...each of the Puranas is encyclopaedic in style, and it is difficult to ascertain when, where, why and by whom these were written: "As they exist today, the Puranas are a stratified literature. Each titled work consists of

material that has grown by numerous accretions in successive historical eras. Thus, no Purana has a single date of composition. It is as if they were libraries to which new volumes have been continuously added, not necessarily at the end of the shelf, but randomly."

Comment 11: l.125 Do ancient texts use the word smoke instead of vapour? It might be interesting, as in the Aristotelian tradition smoke is used for the dry air in opposition to vapour which is used for the moist one;

Response 11: In fact, it is vapour (the moist air). The 'smoke' is mainly related with the burning. However, to symbolize the burning process (here evaporation process), it was termed as smoke. It has been corrected as 'vapour' in the revision.

For enhanced understanding this sentence maybe rectified as: The Vayu Purana (Verse 51. 14-15-16) states that "the water evaporated by sun rises to atmosphere by means of the capillarity of air, and gets cooled and condensed and then it rains".

Comment 12: l.132 Add an English translation (as well for the other citations and titles, see before in the general comment);

Response 12: Thank you for the suggestion. It will be added in the revised manuscript, as suggested.

Comment 13: ll.162—163 It is a very interesting point, as the veins metaphore was common also in other contexts (see e.g. Leonard from Vinci). What feds such veins, as it is reported by Brihat Samhita? And which is the direction along which do they flow?

Response 13: In Brihat Samhita (Chapter 54, Dakargalam), the veins symbolize the 'water table' and the water that falls from the sky feed such veins. It also mentions that the techniques for finding groundwater will be different for different regions and will depend on the type of the landuse and landcover [Verse 54.86]. There are also mentions of the plant species/stone pitching in details for bank protection of water

channel. Here, it would be appropriate to mention Murty (1987) that Varajmihir could be ascertained as the 'earliest hydrologist' of the contemporary world similar to the Leonardo da Vinci, 'Master of Water'.

Comment 14: ll.216—217 probably not necessary;

Response 14: As suggested, this has been removed in the revised manuscript.

Comment 15: l.223 Kautilya. . . : add a reference;

Response 15: The reference Shamasastry, (1961) is added.

Comment 16: l.231 It seems an astronomical approach, rather than an empiristic one: were there found evidences for multiannual precipitation cycles?

Response 16: We agree with the Reviewer. Distinctively, the Arthasastra, does not mention about the multi-annual precipitation cycle; however, it mentions the precipitation cycles based on the types of the 'clouds' as "three are the clouds that continuously rain for seven days; eighty are they that pour minute drops; and sixty are they that appear with the sunshine–this is termed rainfall" (Shamasastry, 1961).

Comment 17: l.242 Please, check whether capillary is properly used;

Response 17: Here, capillary (actual word in Sanskrit is 'NAADI' means artery, column, nerve, pulse) and hence we have replaced it with 'air columns'.

Comment 18: l.257 In which sense it is used change in the direction of flow of ground-water?

Response 18: Thank you for this comment. The sentence "Well before many centuries of Christ" has been replaced with "based on the extensive reviews of the works on water sciences from Mature Harappan civilization to the Mauryan period, it can be established very well that the ancient Indians were aware of cloud formation, rainfall prediction and its measurements, underground water bearing structures, high and low water tables at different places, hot and cold springs, groundwater utilization by means

of wells, well construction methods and equipment, underground water quality and even the artesian well schemes.

Comment 19: l.260 Artesian wells seems not been introduced before, a reference will be useful;

Response 19: It is already mentioned in Line 75.

Comment 20: l.267 In which sense are introduced Eastern and Western hemispheres?

Response 20: Eastern and Western hemispheres represent the 'whole ancient world' (Yannopoulos et al., 2015). Further, the Eastern Hemisphere is sometimes called the "Old World," and the Western Hemisphere is called the "New World." However, the Western Hemisphere is a purely geographic term and should not be confused with other mentions of the "western" world, which is often used to describe parts of Europe, North America and other world regions that share some economic, social, and cultural values (https://www.nationalgeographic.org/encyclopedia/hemisphere/).

Comment 21: ll.281—282 It seems more a saqiya than a naoor / noria: could the Authors add few details?

Response 21: Agree with the views of the Reviewer. 'Asmacakra' was used for lifting water from wells for irrigation purposes. Few more details are further added in the next response.

Comment 22: l.285 and followings Probably it is not necessary to enter here the debate on the origin of the noria, or it is better to strengthen the cited references base on this topic;

Response 22: Thank you for this useful suggestion. We would support the statement with references. During the Vedic period, the water for irrigation purposes was taken from lakes (hrada), canals (kulya), and wells. The exact meaning of the 'asma-cakra' is 'stone-pully'or a 'disk of stone'. The buckets (kosa) tied with the strings made of leather (varatra) were pulled around a stone-pulley and then emptied into the channels

(Mukerji, 1960; Yadav, 2008). Arthasastra mentions irrigating the agricultural fields by raising water from rivers, lakes, tanks and wells using a mechanical device known as 'Udghatam' (Srinivasan, 1970).

Comment 23: l.336 In which sense low cost is used?

Response 23: There are many evidences that the Harappans constructed low cost water harvesting structures using locally available materials through public participation. The Dholavira city is located between the smaller streams Mansar in North and Manhar in South, equipped with series of small check dams, stone drains for diverting water, bunds to reduce the water velocity and thus reduce siltation in the main reservoirs (Eastern and Western Reservoirs) (Nigam et al., 2016; Agrawal et al., 2018). The Gabarbands were also in use in Harappan civilization. Similarly, the Ahar-Pyne system (an excellent example of Participatory Irrigation Management and Rainwater Harvesting in Mauryan Era) are the examples of low-cost sustainable rainwater harvesting structures.

Comment 24: l.340 and followings Rabi irrigation was a spate irrigation, a basin irrigation, or a furrows irrigation?

Response 24: It was mainly Spate irrigation throughout the Indus valley civilization (Miller, 2006; Petrie et al., 2017; Petrie, 2019) in form of Canal, Well and Lift irrigation. In the Indus context, it has been argued that perennial and ephemeral water courses were exploited for flood inundation when present, and when not, the inhabitants relied on rainfall, small-scale irrigation, well/lift irrigation and also ponds to supply water (Miller, 2006; Miller, 2015; Petrie, 2017; Weber, 1991, Petrie and Bates, 2017) and Pyne-Ahar system during the Mauryan era.

Comment 25: l.364 . . . an act of religious merit: it is very interesting to unveil the cultural link between the humans and the Nature. Could the Authors better detail in which sense building reservoirs was considered a religious merit?

[Figure]

Response 25: The religious merit indicates for 'the welfare and well-being of the society'. The Arthasastra mentions that 'He (the King) shall construct reservoirs (sétu) filled with water either perennial or drawn from some other source. Or he may provide with sites, roads, timber, and other necessary things those who construct reservoirs of their own accord. Likewise, in the construction of places of pilgrimage (punyasthána) and of groves. The State control of irrigational activities were great incentive for the agriculturists (Bhattacharya, 2012).

Comment 26: l.379 These dams seems more barrages, eventually used also for spate irrigation. Could the Authors add some more details on the discharge regime and on the use of these dams? Is it a wadi regime?

Response 26: These dams were used for spate irrigation for rice cultivation to support increasing population during the early-historic period (from the 3rd century BC), which seem to be implied by local settlement patterns and indeed the distribution of large monastic sites in Sanchi area.

These dams were specifically built for irrigation purposes, specifically for irrigation of rice (Shaw and Sutcliffe, 2001). According to Shaw and Sutcliffe (2005), it is more likely that the Sanchi reservoirs were part of the complementary irrigation system by providing extensive irrigation for rice cultivation and would have also supplemented rabi crops due to higher moisture holding capacity of the black cotton soils found in that region.

Yes, it is a wadi regime having mainly two perennial (Betwa and Bes) rivers and various nallas (streams). Rainfall is highly seasonal in this area and about 90% of the rainfall occurs in the mid of June to Sept. There is a period of water deficit from January to June (when evapotranspiration exceeds rainfall) followed by a period of July to September (rainfall exceeds evapotranspiration) (Shaw and Sutcliffe, 2001).

Comment 27: l.381 Is the return period referred to present climate or it was estimated for the ancient one?

Response 27: Yes, the return period refers to the present climate.

Comment 28: ll.434—440 Probably not necessary here, and more useful in the Introduction;

Response 28: Agreed. This change will be incorporated in the revised manuscript.

Comment 29: l.447 tapered terra–cotta pipes: Could the Authors add some details on these pipes? They seem frustum–of–cone shaped fistulae common in the Central Asia oases and Latin world;

Response 29: Thank you. We will add details as suggested. The terracotta pipes were used for water supply and sewage, and the sewerage and drainage systems in Harappan civilization (Angelakis and Zheng, 2015). The Terracotta pipes are clay pipes with bell and spigot joints, collars and stop sealed with cement (De Feo et al., 2014). The pipes were built by well-burned bricks (Gray, 1940) having U-shape cross-section and set in clay mortar with various coverings (brick slabs, flagstones or wooden boards) could be removed easily for cleaning the pipes. These ancient terra-cotta pipes, still sound after nearly five thousand years, are the precursor of our modern vitrified clay spigot-and-socket sewer pipe (Gray, 1940).

Several types of stone and terracotta conduits and pipes were also used to transfer water, and drain storm water and wastewater in Minoan Civilization (ca. 3200–1100 BC) (De Feo et al., 2014).

Comment 30: ll.463—465 It sounds not very clear, probably not necessary.

Response 30: This comment will be addressed in the revised version of the manuscript.

References

1. Olson, R. G.: Technology and Science in Ancient Civilizations, ABC-CLIO., 2009.
2. McClellan III, J. E. and Dorn, H.: Science and Technology in World History: An Introduction, JHU Press., 2015.

3. Wittfogel, K. A.: Developmental aspects of hydraulic societies, in Irrigation Civilizations: A Comparative Study, pp. 43–57, Washington DC. [online] Available from: http://www.columbia.edu/itc/anthropology/v3922/pdfs/wittfogel.pdf, 1955.

4. Pruthi, R.: Prehistory and Harappan civilization, APH Publishing., 2004. 5. De Feo, G., Antoniou, G., Fardin, H. F., El-Gohary, F., Zheng, X. Y., Reklaityte, I., Butler, D., Yannopoulos, S. and Angelakis, A. N.: The Historical Development of Sewers Worldwide, Sustainability, 6(6), 3936–3974, doi:10.3390/su6063936, 2014.

6. Jansen, M.: Water supply and sewage disposal at Mohenjo‐Daro, World Archaeology, 21(2), 177–192, doi:10.1080/00438243.1989.9980100, 1989.

7. Jones, D. E.: Urban hydrology-a redirection, Civil Engineering, 37(8), 58-, 1967.

8. Maner, A. W.: Public works in ancient Mesopotamia, Civil Engineering, 36(7), 50–51, 1966.

9. Mays, L. W.: A very brief history of hydraulic technology during antiquity, Environmental Fluid Mechanics, 8(5–6), 471–484, 2008.

10. Fan, C. T.: An ancient draining station was discovered in Old Lingzi city, Space Water Eng, 6, 25, 1987.

11. Khan, S., Dialynas, E., Kasaraneni, V. K. and Angelakis, A. N.: Similarities of Minoan and Indus Valley Hydro-Technologies, Sustainability, 12(12), 4897, doi:10.3390/su12124897, 2020.

12. Mujumdar, P. P. and Jain, S.: Hydrology in Ancient India: Some Fascinating Facets, in EGU General Assembly Conference Abstracts, vol. 20, p. 8690., 2018.

13. Naz, F. and Subramanian, S. V.: Water management across space and time in India, Working Paper, ZEF Working Paper Series. [online] Available from: https://www.econstor.eu/handle/10419/88305 (Accessed 7 July 2020), 2010.

14. Pant, N. and Verma, R. K.: Tanks in Eastern India: A Study in Exploration, IWMI.,

2010.

15. Sengupta, N.: Irrigation: Traditional vs Modern, Economic and Political Weekly, 20(45/47), 1919–1938 [online] Available from: https://www.jstor.org/stable/4375013 (Accessed 7 July 2020), 1985.

16. Verma, N. M. P.: Irrigation in India: Themes on Development, Planning, Performance and Management, M.D. Publications Pvt. Ltd., 1993.

17. O'Malley, L. S. S.: Bengal District Gazetters–Gaya, Superintendent, Government Printing, Bihar and Orissa, Calcutta, 146–147, 1919.

18. Roy Choudhry, P. C.: Bihar District Gazetters, Gaya, Government of Bihar, Patna, 205, 1957.

19. Dimmitt, C. and van Buitenen, J. A. B.: Classical Hindu Mythology: A Reader in the Sanskrit Puranas, Philadelphia: Temple University Press., 1978.

20. Murty, K. S.: Varahamihira, the Earliest Hydrologist, IN: Water for the Future: Hydrology in Perspective. IAHS Publication, (164), 1987.

21. Shamasastry, R.: Kauáź■ilya's ArthaśÄĄstra, Mysore Printing and Publishing House., 1961.

22. Yannopoulos, S. I., Lyberatos, G., Theodossiou, N., Li, W., Valipour, M., Tamburrino, A. and Angelakis, A. N.: Evolution of Water Lifting Devices (Pumps) over the Centuries Worldwide, Water, 7(9), 5031–5060, doi:10.3390/w7095031, 2015.

23. Mukerji, R. K.: Ancient Indian education: Brahmanical and Buddhist, Motilal Banarsidass., 1960.

24. Yadav, A. L.: Some materials for the study of agriculture in Vedic India: Problems and Perspectives., in History of Agriculture in India (upto c.1200 AD), vol. 5, pp. 235–244, Centre for Studies in Civilizations, Delhi, India., 2008.

25. Srinivasan, T. M.: Water Lifting Devices in Ancient India: Their Origin and Mechanisms, Indian journal of History of Science, 5, 379–389 [online] Available from: https://insa.nic.in/writereaddata/UpLoadedFiles/IJHS/Vol05_2_15_TMSrinivasan.pdf, 1970.

26. Nigam, R., Dubey, R., Saraswat, R., Sundaresh, Gaur, A. S. and Loveson, V. J.: Ancient Indians (Harappan settlement) were aware of tsunami/storm protection measures: a new interpretation of thick walls at Dholavira, Gujarat, India, Current Science, 111(12), 2040–2043 [online] Available from: https://www.jstor.org/stable/24911592 (Accessed 27 April 2020), 2016.

27. Agrawal, S., Majumder, M., Bisht, R. S. and Prashant, A.: Archaeological Studies at Dholavira Using GPR, Current Science, 114(04), 879, doi:10.18520/cs/v114/i04/879-887, 2018.

28. Miller, H. M.-L.: Water supply, labor requirements, and land ownership in Indus floodplain agricultural systems, in Agricultural Strategies, pp. 92–128, Cotsen Institute of Archaeology, UCLA, Los Angeles., 2006.

29. Petrie, C. A., Singh, R. N., Bates, J., Dixit, Y., French, C. A. I., Hodell, D. A., Jones, P. J., Lancelotti, C., Lynam, F., Neogi, S., Pandey, A. K., Parikh, D., Pawar, V., Redhouse, D. I. and Singh, D. P.: Adaptation to Variable Environments, Resilience to Climate Change: Investigating Land, Water and Settlement in Indus Northwest India, Current Anthropology, 58(1), 1–30, doi:10.1086/690112, 2017.

30. Petrie, C. A.: Diversity, variability, adaptation and 'fragility' in the Indus Civilization, McDonald Institute for Archaeological Research., 2019.

31. Angelakis, A. N. and Zheng, X. Y.: Evolution of Water Supply, Sanitation, Wastewater, and Stormwater Technologies Globally, Water, 7(2), 455–463, doi:10.3390/w7020455, 2015.

32. Bhattacharya, P. K.: Irrigation and Agriculture In Ancient India. Sectional President's Address, Proceedings of the Indian History Congress, 73, 18–34 [online] Available from: https://www.jstor.org/stable/44156186 (Accessed 27 April 2020), 2012.

33. Gray, H. F.: Sewerage in Ancient and Mediaeval Times, Sewage Works Journal, 12(5), 939–946 [online] Available from: https://www.jstor.org/stable/25029094 (Accessed 8 July 2020), 1940.

34. Miller, H. M. L.: Surplus in the Indus Civilisation, agricultural choices, social relations, political effects, in Surplus: The Politics of Production and the Strategies of Everyday Life., 2015.

35. Petrie, C. A. and Bates, J.: 'Multi-cropping', Intercropping and Adaptation to Variable Environments in Indus South Asia, J World Prehist, 30(2), 81–130, doi:10.1007/s10963-017-9101-z, 2017.

36. Shaw, J. and Sutcliffe, J.: Ancient irrigation works in the Sanchi area: an archaeological and hydrological investigation, South Asian Studies, 17(1), 55–75, doi:10.1080/02666030.2001.9628592, 2001.

37. Shaw, J. and Sutcliffe, J.: Ancient Dams and Buddhist Landscapes in the Sanchi area: New evidence on Irrigation, Land use and Monasticism in Central India, South Asian Studies, 21(1), 1–24, doi:10.1080/02666030.2005.9628641, 2005.

38. Weber, S. A.: Plants And Harappan Subsistence: An Example Of Stability And Change From Rojdi, Oxford and IBH Publishing, New Delhi., 1991.

[Figure]

Latitude: 25.189014
Longitude: 84.210944
Elevation: 90.63m
Accuracy: 3.2m
Time: 23-06-2020 10:52

**Fig. 1.** Figure 1: Renovated Ahar-Pyne system in Bihar.

---

## Referee Comment (RC2) · Anonymous Referee #2 · 10 Jul 2020

The comments on the paper 'Hydrology and Water resources Management in Ancient India' by Pushpendra et al. Authors have made the efforts to bring out the state-of-the art on development of Hydrology and Water Resources in ancient India with reference to mechanism of rainfall and its measurements; Water management Technology and Waste Management Technology. The manuscript is well written and very interesting, which highlight the rich inheritance of India in Water resources management.

While going through the entire manuscript, I could observe that authors have brought out clearly the developments which took place in 'Indus civilization' during 3000 BC to 1500 BC, Vedic period between 1500 BC -500BC and Mauryan dynasty during 400BC to 184 BC.

The following points seem to be missing in the manuscript, though authors have high-
lighted the limitations in deciphering the literature at: Point No. (6) of the Summary and Conclusions.

(a) In the manuscript, I could see the remains of 'water resource Technology' of earliest Harappan/Indus valley civilization are available at present. The description of Vedic period, which came afterwards are given in Vedas (text) only, their physical descriptions are not available at present though they came after Indus civilization. Are such Vedic descriptions pertain to the period much before Indus civilization?

(b) Also, the description of rainfall is available in Ramayana and Mahabharat. However, the period for which such descriptions are given in these literatures is missing. For example, Ramayana was scripted during 200 BC, but its description belongs to which period? Such description will be of much interest to readers from India.

(c) Though the period of Indus valley civilization is mentioned in the literature, however, which ruler ruled that period, is not available. Further, what was the major reasons for collapse of Indus valley civilization? Was it water crisis which led to ruin of entire civilization? The description like Maurya dynasty seems to be more appealing.

---

## Author Comment (AC2) · 16 Jul 2020

Response to the Reviewer's Comments

We thank the reviewer for the constructive comments and suggestions to improve the manuscript. We provide here our responses to the comments and mention the actions to be taken on the manuscript where relevant.

General Comment: The comments on the paper 'Hydrology and Water resources Management in Ancient India' by Pushpendra et al. Authors have made the efforts to bring out the state-of-the art on development of Hydrology and Water Resources in ancient India with reference to mechanism of rainfall and its measurements; Water management Technology and Waste Management Technology. The manuscript is well written

and very interesting, which highlight the rich inheritance of India in Water resources management.

While going through the entire manuscript, I could observe that authors have brought out clearly the developments which took place in 'Indus civilization' during 3000 BC to 1500 BC, Vedic period between 1500 BC -500BC and Mauryan dynasty during 400BC to 184 BC.

Response: Thank you for the positive feedback.

Comment : The following points seem to be missing in the manuscript, though authors have highlighted the limitations in deciphering the literature at: Point No. (6) of the Summary and Conclusions.

Comment 1: In the manuscript, I could see the remains of 'water resource Technology' of earliest Harappan/Indus valley civilization are available at present. The description of Vedic period, which came afterwards are given in Vedas (text) only, their physical descriptions are not available at present though they came after Indus civilization. Are such Vedic descriptions pertain to the period much before Indus civilization?

Response 1: It is mentioned in the manuscript [page # 1; Line #12] that the Vedic Period followed the Indus Valley Civilization (IVC) period. More clearly, after the de-urbanization phase [~1900-1500 BC] of the IVC, the Vedic period came into existence and is generally bracketed between [~1500-500 BC] (Kathayat et al., 2017; Witzel, 2014; Sen, 1999).

Therefore, the beginning of the 'Vedic Period' in India is assumed at about ~1500 BC and the 'Rigveda' (the earliest of the four Vedas) and many other Vedic texts were composed in this period and in later periods (Kathayat et al., 2017; Sen, 1999; Witzel, 1997). With this, the Referee may also take note of the Response 10 C of the comment of Referee 1. [C-10 a] about the periods of the Vedic texts.

Along with this it would be also interesting to quote Kenoyer (2003) : "Our information

is hampered by the fact that most of the Indus settlements dating to the 'Vedic Period' have either been destroyed by later erosion or brick robbing or are covered by continuous inhabitation, which makes excavation impossible". It needs to be noted that surprisingly, both Harappa and Mohenjo Daro also supported later settlements dating to this time, but these levels have been badly disturbed (Kenoyer, 2003).

Chronologically, followed by the fall of the IVC, the Vedic period can be further classified into two stages as : 'Early Vedic Period [∼1500-1100 BC]' and 'Late Vedic Period [∼1100-500 BC]' (Kathayat et al., 2017). Worth mentioning Witzel (1987 & 1999) that 'the Early Vedic period (as attested in the Rigveda hymns) was marked by tribal or pastoral societies, centered in the northern Indus Valley'. However, by the end of this period, the Vedic Society shifted from nomadic life to the settled agriculture with movements towards the east into Gangetic Plains. During the 'Late Vedic Period', the agriculture, metal, commodity production, and trade was largely expanded (Kathayat et al., 2017). After the 'Late Vedic Period' the period of 'Mahajanpadas' came into existence and finally converges into the 'Mauryan Empire'.

As for as the physical description of the 'water resources technologies' is concerned, we have elaborately discussed this in our manuscript at many places, e.g., [Page # 8; Lines: 271-300]. However, it would be appropriate to mention at this juncture that much more research is further needed for 'Vedic Period [1500-500 BC]'on various unexplored aspects of the Vedic Texts from Vedas to Puranas and many other Samhitas [see: Conclusion # 6; Page 14; Line # 502].

Comment 2: Also, the description of rainfall is available in Ramayana and Mahabharat. However, the period for which such descriptions are given in these literatures is missing. For example, Ramayana was scripted during 200 BC, but its description belongs to which period? Such description will be of much interest to readers from India.

Response 2: As observed by Goldman (1984), Brockington (1984, 2000) and Murthy (2003), the core of the epic Ramayana is as old as ∼800-500 BC. The epic Ramayana

is based on the ancient 'ballads/tales' handed down by the 'sutas' (hymns) from generations to generations and compiled between ∼300 BC-200 AD by 'Valmiki' (Winternitz, 1996). Bhargava (1982) also mention that the original portion of the Ramayana was composed by the poet Vãlmíki about a thousand years after the event on the basis of tales handed down by the hymns. The exact composition period is, however, largely differed by many authors (See, Sharma, 1990; Macdonnel, 1919; Keith, 1915). However, this topic is beyond the scope of this study.

Comment 3: Though the period of Indus valley civilization is mentioned in the literature, however, which ruler ruled that period, is not available. Further, what was the major reasons for collapse of Indus valley civilization? Was it water crisis which led to ruin of entire civilization? The description like Maurya dynasty seems to be more appealing.

Response 3: Thank you for this comment, The 'single state' concept was not applicable to the any of the cities of the Indus Valley Civilization, as do we have for the other contemporary civilizations such as Mesopotamia, i.e., the evidence of centralized control— such as the palaces, temples and differentiated burials (Kenoyer, 1994; Possehl, 1998, 2003). The Indus society was based on the shared concepts of power and dominance and the military conquest pattern has not been found in the Indus Valley Civilization (Kenoyer, 2003). However, more information will be available to the world once the linguists are able to decipher the Harappan script as 'inscribed' on the seals, amulets and pottery vessels (Kenoyer, 2003).

Major reasons for collapse of Indus valley civilization (IVC):

Many factors - including climatic, economic and political - have been cited in the past as reasons behind decline of IVC. However, no single explanation can be thought of to be the sole descriptor of this decline. These factors perhaps concatenated to eventually led to the fall of IVC.

Climate Change: The dry epoch that lasted for about 900 years due to weakening of Indian Summer monsoon (around 4350 years ago) adversely impacted the agrarian

society of IVC (Das, 2018; Dixit et al., 2014). The period of long dry spell reduced the snow cover in northwest Himalaya, causing reduced water availability in Indus river (Dutt et al., 2018; Kathayat et al., 2017). The reduction in water availability severely impacted agricultural systems (Sarkar et al., 2016) and production which ultimately lead to the migration of population towards Gangetic plains.

Infectious Diseases: The vulnerable state of Harappan society is compounded by concurrent social and economic changes, promoting further disintegration of IVC. The stratified social structure and urbanization facilitated propagation of infectious diseases (leprosy, tuberculosis) within the marginalized population. These factors led to massive migration of population from Indus Valley around 1900 B.C. (Schug et al., 2013).

Natural Disasters: The presence of silt deposits, topographic and geological anomalies suggest the occurrence of massive floods was related to the decline of IVC. The tectonic disturbances might have altered the course of Indus river affecting the water availability for agricultural production (Dales, 1966).

References:

Bhargava, P.L.: A Fresh Appraisal of the Historicity of Indian Epics. Annals of the Bhandarkar Oriental Research Institute, 63(1/4), pp. 15-28, 1982.

Dales, G. F.: THE DECLINE OF THE HARAPPANS, Scientific American, 214(5), 92–101 [online] Available from: https://www.jstor.org/stable/24930939 (Accessed 11 July 2020), 1966.

Das, B.: A prolonged drought destroyed Indus Valley Civilisation, new study says, Nature India [online] Available from: https://www.natureasia.com/en/nindia/article/10.1038/nindia.2018.61 (Accessed 11 July 2020), 2018.

Dixit, Y., Hodell, D. A. and Petrie, C. A.: Abrupt weakening of the summer monsoon in northwest India ∼4100 yr ago, Geology, 42(4), 339–342, doi:10.1130/G35236.1, 2014.

Dutt, S., Gupta, A. K., Wünnemann, B. and Yan, D.: A long arid interlude in the Indian summer monsoon during âĹij4,350 to 3,450 cal. yr BP contemporaneous to displacement of the Indus valley civilization, Quaternary International, 482, 83–92, doi:10.1016/j.quaint.2018.04.005, 2018.

Kathayat, G., Cheng, H., Sinha, A., Yi, L., Li, X., Zhang, H., Li, H., Ning, Y. and Edwards, R. L.: The Indian monsoon variability and civilization changes in the Indian subcontinent, Science Advances, 3(12), e1701296, doi:10.1126/sciadv.1701296, 2017.

Keith, A.B.: The Date of the Ramayana." Journal of the Royal Asiatic Society. pp. 318-321., 1915,

Kenoyer, J. M.: The Harappan state: was it or wasn't it, Madison, WI: Prehistory Press., 1994.

Kenoyer, J. M.: Uncovering the keys to the lost Indus cities, Scientific American, 289(1), 66–75, 2003.

Macdonell, A.A.: Ramayana." Encyclopedia of Religion and Ethics, Vol- 10, edited by James Hastings, Edinburgh: T. & T. Clark, pp. 574-578, 1919.

Murthy, S.S.N.: A Note on Ramayana. Electronic Journal of Vedic Studies, 10(6), pp. pp. 1-18. © ISSN 1084 -7561, 2003.

Possehl, G. L.: Sociocultural complexity without the State. The Indus Civilization, in Archaic states, vol. School of American Research advanced seminar series, edited by G. M. Feinman and J. Marcus, pp. 261–291, School of American Research Press, Santa Fe, N.M., 1998.

Possehl, G. L.: The Indus Civilization: an introduction to environment, subsistence, and cultural history, in Indus ethnobiology, edited by S. Weber and W. Belcher, pp. 1–20., 2003.

Sarkar, A., Mukherjee, A. D., Bera, M. K., Das, B., Juyal, N., Morthekai, P., Deshpande,

R. D., Shinde, V. S. and Rao, L. S.: Oxygen isotope in archaeological bioapatites from India: Implications to climate change and decline of Bronze Age Harappan civilization, Scientific Reports, 6(1), 26555, doi:10.1038/srep26555, 2016.

Schug, G. R., Blevins, K. E., Cox, B., Gray, K. and Mushrif-Tripathy, V.: Infection, Disease, and Biosocial Processes at the End of the Indus Civilization, PLOS ONE, 8(12), e84814, doi:10.1371/journal.pone.0084814, 2013.

Sen, S. N.: Ancient Indian history and civilization, New Age International., 1999.

Winternitz, M. A.: History of Indian Literature, Motilal Banarsidas, Delhi 1996 (reprint).

Witzel, M.: Central Asian roots and acculturation in South Asia: linguistic and archaeological evidence from Western Central Asia, the Hindukush and northwestern South Asia for early Indo-Aryan language and religion, in Liguistics, archaeology and the human past, edited by T. Osada, pp. 87–211., 2005.

Witzel, M.: The Development of the Vedic Canon and its Schools : The Social and Political Milieu", Harvard University. pp. 261–264, 1997.

---

## Author Response (AR1)

Journal: Hydrology Earth System Sciences
Title: Hydrology and Water Resources Management in Ancient India
Authors: Pushpendra Kumar Singh, Pankaj Dey, Sharad Kumar Jain, Pradeep Mujumdar
Manuscript No: hess-2020-213

Dear Editor, Associate Editor, Dr. Stefano Barontini, and Anonymous Reviewer

We thank the Editor, Prof. Roberto Ranzi, Associate Editor and two reviewers for providing comments and suggestions for overall improvement in the structure and content of the manuscript. We have addressed the comments of Dr. Stefano Barontini and the anonymous reviewer. Our detailed response to reviewers' comments is included in the following pages, with our responses shown in blue. We hereby submit the revised version of the manuscript.

On behalf of all co-authors-
Sincerely,

P P Mujumdar
(Corresponding author)
August 2020.

**Comments from Dr. Prof. Stefano Barontini**

I read with interest the contribution Hydrology and water resources management in ancient India by Singh et al., in which, on the basis of an accurate bibliographical review, the Authors present many aspects of the multifaceted hydrological and hydraulic knowledge in ancient India. The themes addressed are the comprehension of the hydrological cycle, the precipitation measurements, the water management (with more evidence to the hydraulic structures than to the management practices) and the wastewater management.

The paper is well written and thoroughly argued, and it makes a state of the art of the matter, provided that the topic stands between many disciplines (history, archaeology, hydraulic engineering, history of technology and history of culture). Therefore, the paper might be eventually recommended for publication, but I encourage the Authors to strengthen its unitary perspective, in order to depict a wide portrayal, thus avoiding the risk of giving the idea of a collection of cases.

We thank Prof. Stefano for the positive feedback and for offering several comments to improve the manuscript. We have greatly benefited by the comments of the reviewer. We provide here our responses and mention how we would modify the manuscript.

Comment 1: As a first point, for example, it might be useful to explicitly state both in the Abstract and in the Introduction which are the geographical and historical boundaries of the matters, and possibly why these boundaries were chosen, and the aims and the methods of the research.

Response 1: The geographical region covers the entire Indian sub-continent to the east of the Indus river. It includes the parts of the Harappan civilization (in the present-day Pakistan) and entire India. These boundaries encompass the major centers/regions of the development in the ancient India. It would be appropriate to quote Olson (2009) here: 'India was not re-united for nearly 500 years after the collapse of the Mauryan Empire, so its end forms a logical place to end the discussion of the ancient India'. Our discussion in the manuscript is mainly concerned with this period.

This has been incorporated in the revised version of the manuscript (Abstract: Page # 1; Lines: 28-33 and in Introduction (Section 1) at Page # 4; Lines: 175-182).

Comment 2: Much information presented in the Introduction might be effectively contextualized in the following sections, whereas in the Introduction it is recommended to declare which is the order along which the matter is presented in each section (e.g. historical order, or process– or technology–based order, etc.).

Response 2: The manuscript has been prepared in view of the process-or technology- based order. While doing so, the historical order of those processes or technologies has also been maintained in the manuscript. The text of the manuscript in each section (including the Introduction) has been re-structured accordingly. See, e.g., in Introduction (1st Harappan: Lines: 49-73; Vedic Period: Lines: 75-94; Mauryan Empire: Lines 96-142). Similar pattern can be observed in rest of the sections of the manuscript.

Comment 3: Also the concept of "hydraulic civilization", which is sometimes used in the paper, might be better defined in the Introduction. In fact in all the ancient and modern societies the water management plays a crucial role, but the attribute of "hydraulic civilization" is nowadays preferably used to identify those civilizations which survival was deeply linked with the capability of managing the water–related issues (as e.g. the water scarcity, the soil salinization, or the floods) and, in most of the cases, the management was centralized via well-structured groups of technicians and skilled workers (as it was e.g. the case of the great Central Asia oases).

Response 3: Thank you for this insightful comment. In this manuscript, the concept of hydraulic civilization was referred to in respect of the Harrapa civilization and the Mauryan Empire (321-297 BC) in India. Following discussion has been added appropriately in the revised manuscript in the Introduction Section [Lines: 108-139].

"According to McClellan III and Dorn (2006), the Mauryan Empire was 'first and foremost a great hydraulic civilization.' Megasthenes (A Greek traveller in Chandragupta's Court, around 300 BC), mentions that 'more than half of the arable land was irrigated and in agriculture and produced two harvests in a year'. Further, there was a special department for supervision, construction and maintenance of a well-developed irrigation system with extensive canals and sluices, wells, lakes and tanks. The same bureau was responsible for planning and settlement of the uncultivated land. A similar description of the different institutional arrangements during Mauryan period can be had from *Arthasastra*. The importance of the hydraulic structures in the Mauryan period can be adjudged on the basis of the punishments/fines to the offenders. As mentioned in the *Arthasastra*, 'When a person breaks the dam of a tank full of water, he shall be drowned in the very tank; of a tank without water, he shall be punished with the highest amercement; and of a tank which is in ruins owing to neglect, he shall be punished with the middle-most amercement'.

Remarkably, the Mauryan Empire did not lack the other hallmarks associated with the hydraulic civilizations (McClellan III and Dorn, 2006). It had the departments concerned with the rivers, excavating and irrigation along with a number of regional and other superintendents such as the superintendent of rivers, agriculture, weights and measures, store-house, space and time, ferries, boats, and ships, towns, pasture grounds, road-cess, and many others along with many strata of the associated officers such as head of the departments (adhyakshah), collector-general (samahartri), and chamberlain (sannidhatri), etc. Olson (2009) also mentions that there was an extensive irrigation network organised by a state bureaucracy. According to Wittfogel (1955), the Mauryan Empire had virtually all of those characteristics that a hydraulic civilization must possess (though it was late and short lived).

Water pricing was very well defined in the Mauryan Empire. According to *Arthasastra*, those who cultivate irrigating by manual labour (hastaprávartimam) shall pay 1/5th of the produce as water-rate (udakabhágam); by carrying water on shoulders (skandhaprávartimam) 1/4th of the produce; by water-lifts (srotoyantraprávartimam), 1/3rd of the produce; and by raising water from rivers, lakes, tanks, and wells (nadisarastatákakúpodghátam),1/3rd or 1/4th of the produce. The Superintendent of the Agriculture was responsible for compiling the meteorological statistics by using a rain gauge and for observing the sowing of the wet crops, winter crops or summer crops depending on the availability of the water."

Comment 4: Finally, I encourage the Authors to enlighten, on the basis of the investigated literature, the links between the Indian hydraulic culture and that of the surrounding cultures, particularly regarding the water technologies (see below for details).

Response 4: A separate section, [Section 6; Lines: 695-766] with appropriate editing has been added in the revised manuscript.

Hydraulic Inter-linkages between the Ancient Indian and Nearby Cultures

All the ancient civilizations, i.e., Harappan, Egyptian, Mesopotamian, Chinese, and including the Minoan civilization that flourished and attained their pinnacle were largely dependent on degree/extent of their advancements in the field of water technologies. With the efficient management of water resources, they were able to produce more food grains and mitigate the damages due to natural hazards such as droughts and floods. At the same time, the advanced wastewater management techniques helped in healthy lifestyles, hygiene, and clean environments.

The ancient Indian literature, starting from the Harappan civilization to the Vedic Period followed by the Mauryan Empire, the Vedic Samhitas and Puranas, contains detailed discourses on the various processes of hydrological cycle, including groundwater exploration, water quality, well construction, irrigation by channels (kulya). Water technological advancements coupled with the architectural sophistication during the Harappan civilization were at their zenith. Nowhere in the world we had such sophisticated and impressive planning relating to the water supply and effluent disposal system (Jansen, 1989). Almost all houses were having their private wells with bath and toilet area lined with the standard size burnt bricks and draining into the soak pit or into the street drains.

Multiple flushing lavatories attached to a sophisticated sewage system were located in the ancient cities of Harappa and Mohenjo-Daro civilization (Pruthi, 2004). The Great Bath at Mohenjo-Daro and 16 reservoir system of the Dholavira and the Dock yard are the perfect examples of the excellent hydraulic engineering in the Harappan civilization. The Mauryan Empire was named as the 'hydraulic civilization' due to developments of the advanced means of irrigation, construction of wells, dams and reservoirs, rainfall measurements, protection of hydraulic structures, and water pricing systems in place and a stratified establishment of the bureaucratic and engineering establishment.

The effluent disposal drainage systems were well-known to almost all the civilizations at that time with varying level of technological advancements. The Egyptian civilization (~2000-500 BC), lacked the flushing lavatories and sophisticated sewer and wastewater disposal systems at that time as was prevalent in Harappan. The copper pipes were in use in some Pyramids for building bathrooms and sewerage system (De Feo et al., 2014). The Mesopotamian civilization (ca. 4000–2500 BC) also had well-constructed storm drainage and sanitary sewer systems. However, there seems no system of vertical water supply by means of wells and it was even practically unknown in the early urban cultures (Jansen, 1989; De Feo, 2014). According to Jansen (1989) and De Feo et al., (2014), the very efficient drainage and sewerage systems, flushing toilets, which can be compared to the modern ones, re-established in Europe and North America in a century and half ago.

The Mohenjo-Daro city was serviced by at least 700 wells, whereas, the contemporary Egyptian and Mesopotamians had to fetch water bucket-by-bucket from the river and then store in the tanks at homes (Jansen, 1989). The bathing platforms in the Harappan civilizations were also unique as compared to the Mesopotamian and other civilizations. The ancient cities of the Mesopotamian civilization, i.e., UR and Babylone had effective drainage system for storm water control, sewers and drains for household waste and drains specifically for surface runoff (Jones, 1967; Maner, 1966). The ancient Mesopotamians had also developed canal irrigated agriculture and constructed dams across the Tigris river for diverting water to meet the irrigation and domestic supplies. The 'qanat' were widely used in Mesopotamian civilization for transferring the water from one place to another using the gravity. The urban centers of the Sumer (Sumerian) and Akkud (Akkadian) (third millennium BC) had water supplies by canal(s) connected to the Euphrates River. However, this lacks the advancements as compared to the Harappan civilization. The water lifting device were also used in Mesopotamian Civilization and the Saaqia (or water wheel) was widely used for lift irrigation using oxen for irrigating the summer crops (Mays, 2008).The 'asmacakra' and 'Ghatayantra' were widely in use during the Vedic and Mauryan Period. The 'Varshaman' was widely used in Mauryan Empire for rainfall measurements. It may be noted that we do not have any reference of 'rainfall measurement' in other contemporary civilizations in the old world. The Pynes-Ahar system of participatory irrigation and rainwater harvesting is a unique system developed in Ancient India.

In Chinese (Hwang-Ho) civilization, the Shang dynasty (1520-1030 BC) developed extensive irrigation works for rice cultivation. Various water works such as dikes, dams, canals and artificial lakes proliferated across the Chinese civilization. Yu the Great, is acclaimed in China as the 'controller of the waters'. During the period 1100-221 BC, the Lingzi city (covering an area of 15 km2) also had a complex water supply and drainage system, combined with the river, drainage raceway, pipeline and moat (De Feo, et al., 2014). The moat surrounding the town halls had supplies from the river works as daily water uses. The water-fortification (audaka) around the forts was also a prime requirement in the Mauryan Empire. Notably, the drainage system of the Lingzi town is supposed to be the oldest and biggest in the ancient China (Fan, 1987). The drainage systems to collect rainwater and wastewater into pools and finally discharge into river were made of the earthenware pipes. The underground urban drainage systems were also in existence in Chine during the Shan Dynasty (~10-15 BC).

The Minoan civilization (~3200-1100 BC) is considered to be the first and the most important European culture (Khan et al., 2020). The Crete island was the center of the Minoan civilization and was known for architectural and hydraulic operation of its water supply, sewerage, and drainage systems (Khan et al., 2020). Aqueducts made of terracotta were in use for transporting water from the mountain springs. Water cistern were used for storing rainwater and spring water for further transporting it by using aqueduct. Lavatories with the flushing system were also in use in this civilization. In words of Jansen (1989), 'for the first time in the history of mankind, the waterworks developed in Harappan civilization were to such a perfection which was to remain unsurpassed until the coming of the Romans and the flowering of civil engineering and architecture in classical antiquity, more than 2,000 years later'.

Overall, if we closely look at the scale of the hydro-technologies in all the civilizations, the Harappan civilization is not only credited with the more advanced and larger scale application of hydro-technologies (hydrologic, hydraulic and hydro-mechanical) but also worked as a 'landmark' for the contemporary civilizations to achieve the great heights in human civilizations, on the whole.

Comment 5: As a general typographical aspect, I recommend to check and uniform all the emphases and the citations, and to add a complete English translation to all the book titles (the first time they are introduced) and to all the ancient citations.

Response 5: Yes, the suggestions have been taken care of in the revised version of the manuscript.

Comment 6: line 49 add a reference for the citation;

Response 6: The reference, Mujumdar and Jain (2018) has been added in the revised manuscript.

Comment 7: l.53 emphasize variyantra and better detail its functioning;

Response 7: A revised sentence is given here, which will be added in the manuscript: The variyantra (water machine) was similar to the water cooler. According to Megasthenes (an ancient Greek historian in the court of King Chandragupta Maurya), the variyanytra was used by the wealthier sections of the society for cooling the air. This has been added in the revised manuscript [Lines: 99-104].

Comment 8: l.57 pynes and ahars are very interesting structures, also in this case I recommend to better define their functioning (e.g. whether ahars are fed by pynes or by the slopes) and, if possible, their diffusion;

Response 8: Thank you for the suggestion. The Pynes are man-made channels to utilize the river water flowing through the hilly rivers of South Bihar and Chhota Nagpur plateau, whereas the Ahars are catchments with embankments on three sides to store rainwater and the water from the Pynes (Naz and Subramanian, 2010). The Ahar-Pyne system is still widely practiced in these regions and it is a shining example of participatory irrigation management (Pant and Verma, 2010). The Pynes feed many Ahars and several distributaries are then constructed from both Pynes and Ahars for irrigating the field (Sengupta, 1985; Verma, 1993). The Ahar-Pyne system is extremely suitable for the regions having scanty rainfall, highly undulating and rocky terrain, soils with heavy clay or loose sand (lower moisture holding capacity) and steep slope thus causing extensive surface runoff.

The Pynes are of different sizes. If the Pynes are originating from the Ahars, then these are smaller in size (3 to 5 km) and used for irrigating cultivable fields, where as if these originating from the rivers, then the size may vary from 16 to 32 km in length and some of them known as dasianpynes (pynes with 10 branches) to irrigate many thousand acres of the land (O' Malley, 1919). Apart from participatory irrigation system, the Ahar-Pyne system also works as flood mitigation system (Roy Choudhry, 1957). Worth mentioning, recently the Government of Bihar has started the 'renovation' of the traditional water bodies (Ahar-Pyne system) under 'Jal Jeevan Hariyali' programme. This reflects the importance of this ancient hydraulic structure for water harvesting even in the modern times in India (as shown in Figure 1). A brief discussion on this has been added in the manuscript [Lines: 531-548].

Comment 9: l.73 it is meant the Arthashastra of l.50, isn't it?

Response 9: Thank you. Yes, it is same as in line 50, i.e., (Arthashastra). The meaning of the Arthashastra is the 'the science of material gain'.

Comment 10: l.115 it can be inferred. . . : this is an important point for the comprehension of the hydrological cycle.

a.        Since what it is reported, it seems that the correct comprehension of the hydrological cycle was already achieved in ancient India, as it was few centuries later in ancient Greece, before the Aristotelian statement according to which the water of great rivers could not be stored inside the Earth. Are there explicit references to issues related to the infiltration and to the storage in subsoil reservoirs?

b.        This conjecture (the Aristotelian one) paved the way to an (uncorrect) description of the hydrological cycle based on the concurrence of two cycles: one external to the Earth, driven by the Sun, and a more important one internal to the Earth, driven by an engine placed within the Earth's depths. At Authors' knowledge, are there reflections of this conjecture in the Indian late–antiquity hydrological culture?

c.        Moreover, Puranas are reported to be written between 600 B.C. and 700 A.D.: is it possible to provide a closer time range for the ones which are cited by the Authors (and particularly for the Vayu Purana)?

Response 10:

a.        The infiltration process and sub-soil reservoirs is defined in the Brihat Samhita (550 AD) as given in Line # 162-163. However, the Verses 184.15-17 of Mahabharata state that the plants drink water through their roots. It is said that the water uptake process is facilitated by the conjunction of air.
b.        The 'Sun' is the main source of the hydrologic cycle [Lines: 107-108; Page# 3] was very well know from the days of Vedic periods. In Rigveda [Lines: 100-101; Page #3 of the manuscript], it is mentioned therein that 'the God has created ''Sun' and has placed it in such a position……..".
c.        The Puranas are a class of literary texts, all written in Sanskrit verse, whose composition dates from the 4th century BCE to about 1,000 A.D (http://southasia.ucla.edu/religions/texts/puranas/).
Further it would be interesting to quote Dimmitt and van Buitenen (1978): "…each of the Puranas is encyclopaedic in style, and it is difficult to ascertain when, where, why and by whom these were written: "As they exist today, the Puranas are a stratified literature. Each titled work consists of material that has grown by numerous accretions in successive historical eras. Thus, no Purana has a single date of composition. It is as if they were libraries to which new volumes have been continuously added, not necessarily at the end of the shelf, but randomly."

Comment 11: l.125 Do ancient texts use the word smoke instead of vapour? It might be interesting, as in the Aristotelian tradition smoke is used for the dry air in opposition to vapour which is used for the moist one;

Response 11: In fact, it is vapour (the moist air). The 'smoke' is mainly related with the burning. However, to symbolize the burning process (here evaporation process), it was termed as smoke. It has been corrected as 'vapour' in the revision. For enhanced understanding this sentence has been rectified in the revised manuscript. The Vayu Purana (Verse 51. 14-15-16) states that "the water evaporated by sun rises to atmosphere by means of the capillarity of air, and gets cooled and condensed and then it rains".

Comment 12: l.132 Add an English translation (as well for the other citations and titles, see before in the general comment);

Response 12: Thank you for the suggestion. This is added in the revised manuscript, as suggested.

Comment 13: ll.162—163 It is a very interesting point, as the veins metaphore was common also in other contexts (see e.g. Leonard from Vinci). What feds such veins, as it is reported by Brihat Samhita? And which is the direction along which do they flow?

Response 13: In *Brihat Samhita* (Chapter 54, Dakargalam), the veins symbolize the 'water table' and the water that falls from the sky feed such veins. It also mentions that the techniques for finding groundwater will be different for different regions and will depend on the type of the landuse and landcover [Verse 54.86]. There are also mentions of the plant species/stone pitching in details for bank protection of water channel. Here, it would be appropriate to mention Murty (1987) that Varajmihir could be ascertained as the 'earliest hydrologist' of the contemporary world similar to the Leonardo da Vinci, 'Master of Water'. This has been appropriately added in the revised manuscript [Lines: 265-285].

Comment 14: ll.216—217 probably not necessary;

Response 14: As suggested, this has been removed in the revised manuscript.

Comment 15: l.223 Kautilya. . . : add a reference;

Response 15: The reference Shamasastry, (1961) is added.

Comment 16: l.231 It seems an astronomical approach, rather than an empiristic one: were there found evidences for multiannual precipitation cycles?

Response 16: We agree with the Reviewer. Distinctively, the *Arthasastra*, does not mention about the multi-annual precipitation cycle; however, it mentions the precipitation cycles based on the types of the 'clouds' as "three are the clouds that continuously rain for seven days; eighty are they that pour minute drops; and sixty are they that appear with the sunshine--this is termed rainfall" (Shamasastry, 1961).

Comment 17: l.242 Please, check whether capillary is properly used;

Response 17: Here, capillary (actual word in Sanskrit is 'NAADI' means artery, column, nerve, pulse) and hence we have replaced it with 'air columns'. This has been added in the revised manuscript.

Comment 18: l.257 In which sense it is used change in the direction of flow of groundwater?

Response 18: Thank you for this comment. The sentence "Well before many centuries of Christ" has been replaced with "based on the extensive reviews of the works on water sciences from Mature Harappan civilization to the Mauryan period, it can be established very well that the ancient Indians were aware of cloud formation, rainfall prediction and its measurements, underground water bearing structures, high and low water tables at different places, hot and cold springs, groundwater utilization by means of wells, well construction methods and equipment, underground water quality and even the artesian well schemes.

Comment 19: l.260 Artesian wells seems not been introduced before, a reference will be useful;

Response 19: It is already mentioned in Line 89-90 in the revised manuscript.

Comment 20: l.267 In which sense are introduced Eastern and Western hemispheres?

Response 20: Eastern and Western hemispheres represent the 'whole ancient world' (Yannopoulos et al., 2015). Further, the Eastern Hemisphere is sometimes called the "Old World," and the Western Hemisphere is called the "New World." However, the Western Hemisphere is a purely geographic term and should not be confused with other mentions of the "western" world, which is often used to describe parts of Europe, North America and other world regions that share some economic, social, and cultural values (https://www.nationalgeographic.org/encyclopedia/hemisphere/).

Comment 21: ll.281—282 It seems more a saqiya than a naoor / noria: could the Authors add few details?

Response 21: Agree with the views of the Reviewer. 'Asmacakra' was used for lifting water from wells for irrigation purposes. Few more details are further added in the next response.

Comment 22: l.285 and followings Probably it is not necessary to enter here the debate on the origin of the noria, or it is better to strengthen the cited references base on this topic;

Response 22: Thank you for this useful suggestion. We would support the statement with references. During the Vedic period, the water for irrigation purposes was taken from lakes (hrada), canals (kulya), and wells. The exact meaning of the 'asma-cakra' is 'stone-pully'or a 'disk of stone'. The buckets (kosa) tied with the strings made of leather (varatra) were pulled around a stone-pulley and then emptied into the channels (Mukerji, 1960; Yadav, 2008). *Arthasastra* mentions irrigating the agricultural fields by raising water from rivers, lakes, tanks and wells using a mechanical device known as 'Udghatam' (Srinivasan, 1970). This has been added in the revised manuscript [Lines: 478-491].

Comment 23: l.336 In which sense low cost is used?

Response 23: There are many evidences that the Harappans constructed low cost water harvesting structures using locally available materials through public participation. The Dholavira city is located between the smaller streams Mansar in North and Manhar in South, equipped with series of small check dams, stone drains for diverting water, bunds to reduce the water velocity and thus reduce siltation in the main reservoirs (Eastern and Western Reservoirs) (Nigam et al., 2016; Agrawal et al., 2018). The Gabarbands were also in use in Harappan civilization. Similarly, the Ahar-Pyne system (an excellent example of Participatory Irrigation Management and Rainwater Harvesting in Mauryan Era) are the examples of low-cost sustainable rainwater harvesting structures. The lines 445-468 and Ahar-Pyne System in different sections of the revised manuscript.

Comment 24: l.340 and followings Rabi irrigation was a spate irrigation, a basin irrigation, or a furrows irrigation?

Response 24: It was mainly Spate irrigation throughout the Indus valley civilization (Miller, 2006; Petrie et al., 2017; Petrie, 2019) in form of Canal, Well and Lift irrigation. In the Indus context, it has been argued that perennial and ephemeral water courses were exploited for flood inundation when present, and when not, the inhabitants relied on rainfall, small-scale irrigation, well/lift irrigation and also ponds to supply water (Miller, 2006; Miller, 2015; Petrie, 2017; Weber, 1991, Petrie and Bates, 2017) and Pyne-Ahar system during the Mauryan era. The lines 462-468 is added based on this discussion.

Comment 25: l.364 . . . an act of religious merit: it is very interesting to unveil the cultural link between the humans and the Nature. Could the Authors better detail in which sense building reservoirs was considered a religious merit?

Response 25: The religious merit indicates for 'the welfare and well-being of the society'. The *Arthasastra* mentions that 'He (the King) shall construct reservoirs (sétu) filled with water either perennial or drawn from some other source. Or he may provide with sites, roads, timber, and other necessary things those who construct reservoirs of their own accord. Likewise, in the construction of places of pilgrimage (punyasthána) and of groves. The State control of irrigational activities were great incentive for the agriculturists (Bhattacharya, 2012). This has been discussed in lines 550-563 in the revised manuscript.

Comment 26: l.379 These dams seems more barrages, eventually used also for spate irrigation. Could the Authors add some more details on the discharge regime and on the use of these dams? Is it a wadi regime?

Response 26: These dams were used for spate irrigation for rice cultivation to support increasing population during the early-historic period (from the 3rd century BC), which seem to be implied by local settlement patterns and indeed the distribution of large monastic sites in Sanchi area.

These dams were specifically built for irrigation purposes, specifically for irrigation of rice (Shaw and Sutcliffe, 2001). According to Shaw and Sutcliffe (2005), it is more likely that the Sanchi reservoirs were part of the complementary irrigation system by providing extensive irrigation for rice cultivation and would have also supplemented rabi crops due to higher moisture holding capacity of the black cotton soils found in that region.

Yes, it is a wadi regime having mainly two perennial (Betwa and Bes) rivers and various nallas (streams). Rainfall is highly seasonal in this area and about 90% of the rainfall occurs in the mid of June to Sept. There is a period of water deficit from January to June (when evapotranspiration exceeds rainfall) followed by a period of July to September (rainfall exceeds evapotranspiration) (Shaw and Sutcliffe, 2001). This has been discussed in the revised manuscript [Lines: 564-590].

Comment 27: l.381 Is the return period referred to present climate or it was estimated for the ancient one?

Response 27: Yes, the return period refers to the present climate.

Comment 28: ll.434—440 Probably not necessary here, and more useful in the Introduction;

Response 28: Agreed. This change will be incorporated in the revised manuscript.

Comment 29: l.447 tapered terra–cotta pipes: Could the Authors add some details on these pipes? They seem frustum–of–cone shaped fistulae common in the Central Asia oases and Latin world;

Response 29: Thank you. We will add details as suggested. The terracotta pipes were used for water supply and sewage, and the sewerage and drainage systems in Harappan civilization (Angelakis and Zheng, 2015). The Terracotta pipes are clay pipes with bell and spigot joints, collars and stop sealed with cement (De Feo et al., 2014). The pipes were built by well-burned bricks (Gray, 1940) having U-shape cross-section and set in clay mortar with various coverings (brick slabs, flagstones or wooden boards) could be removed easily for cleaning the pipes. These ancient terra-cotta pipes, still sound after nearly five thousand years, are the precursor of our modern vitrified clay spigot-and-socket sewer pipe (Gray, 1940).

Several types of stone and terracotta conduits and pipes were also used to transfer water, and drain storm water and wastewater in Minoan Civilization (ca. 3200–1100 BC) (De Feo et al., 2014). This has been discussed in the revised manuscript [Lines: 653-659].

Comment 30: ll.463—465 It sounds not very clear, probably not necessary.

Response 30: These lines have been deleted in the revised manuscript.

**Comments from Reviewer 2**

The comments on the paper 'Hydrology and Water resources Management in Ancient India' by Pushpendra et al. Authors have made the efforts to bring out the state-of-the art on development of Hydrology and Water Resources in ancient India with reference to mechanism of rainfall and its measurements; Water management Technology and Waste Management Technology. The manuscript is well written and very interesting, which highlight the rich inheritance of India in Water resources management.

While going through the entire manuscript, I could observe that authors have brought out clearly the developments which took place in 'Indus civilization' during 3000 BC to 1500 BC, Vedic period between 1500 BC -500BC and Mauryan dynasty during 400BC to 184 BC.

The following points seem to be missing in the manuscript, though authors have highlighted the limitations in deciphering the literature at: Point No. (6) of the Summary and Conclusions.

We thank the reviewer for the constructive comments and suggestions to improve the manuscript. We provide here our responses to the comments and mention the actions to be taken on the manuscript where relevant.

Comment 1: In the manuscript, I could see the remains of 'water resource Technology' of earliest Harappan/Indus valley civilization are available at present. The description of Vedic period, which came afterwards are given in Vedas (text) only, their physical descriptions are not available at present though they came after Indus civilization. Are such Vedic descriptions pertain to the period much before Indus civilization?

Response 1: It is mentioned in the manuscript [page # 1; Line: 12] that the Vedic Period followed the Indus Valley Civilization (IVC) period. More clearly, after the deurbanization phase [~1900-1500 BC] of the IVC, the Vedic period came into existence and is generally bracketed between [~1500-500 BC] (Kathayat et al., 2017; Witzel, 2014; Sen, 1999).

Therefore, the beginning of the 'Vedic Period' in India is assumed at about _1500 BC and the 'Rigveda' (the earliest of the four Vedas) and many other Vedic texts were composed in this period and in later periods (Kathayat et al., 2017; Sen, 1999; Witzel, 1997). With this, the Referee may also take note of the Response 10 C of the comment of Referee 1. [C-10 a] about the periods of the Vedic texts.

Along with this it would be also interesting to quote Kenoyer (2003) : "Our information is hampered by the fact that most of the Indus settlements dating to the 'Vedic Period' have either been destroyed by later erosion or brick robbing or are covered by continuous inhabitation, which makes excavation impossible". It needs to be noted that surprisingly, both Harappa and Mohenjo Daro also supported later settlements dating to this time, but these levels have been badly disturbed (Kenoyer, 2003). Chronologically, followed by the fall of the IVC, the Vedic period can be further classified into two stages as : 'Early Vedic Period [1500-1100 BC]' and 'Late Vedic Period [1100-500 BC]' (Kathayat et al., 2017). Worth mentioning Witzel (1987 & 1999) that 'the Early Vedic period (as attested in the Rigveda hymns) was marked by tribal or pastoral societies, centered in the northern Indus Valley'. However, by the end of this period, the Vedic Society shifted from nomadic life to the settled agriculture with movements towards the east into Gangetic Plains. During the 'Late Vedic Period', the agriculture, metal, commodity production, and trade was largely expanded (Kathayat et al., 2017). After the 'Late Vedic Period' the period of 'Mahajanpadas' came into existence and finally converges into the 'Mauryan Empire'. This has been addressed in [Introduction: Lines: 75-82]

As for as the physical description of the 'water resources technologies' is concerned, we have elaborately discussed this in our manuscript at many places, e.g., [Page #8; Lines: 271-300]. However, it would be appropriate to mention at this juncture that much more research is further needed for 'Vedic

Period [1500-500 BC]'on various unexplored aspects of the Vedic Texts from Vedas to Puranas and many other Samhitas [Lines: 504-510].

Comment 2: Also, the description of rainfall is available in Ramayana and Mahabharat. However, the period for which such descriptions are given in these literatures is missing. For example, Ramayana was scripted during 200 BC, but its description belongs to which period? Such description will be of much interest to readers from India.

Response 2: As observed by Goldman (1984), Brockington (1984, 2000) and Murthy (2003), the core of the epic Ramayana is as old as ~800-500 BC. The epic Ramayana is based on the ancient 'ballads/tales' handed down by the 'sutas' (hymns) from generations to generations and compiled between ~300 BC-200 AD by 'Valmiki' (Winternitz, 1996). Bhargava (1982) also mention that the original portion of the Ramayana was composed by the poet Vãlmíki about a thousand years after the event on the basis of tales handed down by the hymns. The exact composition period is, however, largely differed by many authors (See, Sharma, 1990; Macdonnel, 1919; Keith, 1915). However, this topic is beyond the scope of this study.

Comment 3: Though the period of Indus valley civilization is mentioned in the literature, however, which ruler ruled that period, is not available. Further, what was the major reasons for collapse of Indus valley civilization? Was it water crisis which led to ruin of entire civilization? The description like Maurya dynasty seems to be more appealing.

Response 3: Thank you for this comment, The 'single state' concept was not applicable to the any of the cities of the Indus Valley Civilization, as do we have for the other contemporary civilizations such as Mesopotamia, i.e., the evidence of centralized control - such as the palaces, temples and differentiated burials (Kenoyer, 1994; Possehl, 1998, 2003). The Indus society was based on the shared concepts of power and dominance and the military conquest pattern has not been found in the Indus Valley Civilization (Kenoyer, 2003). However, more information will be available to the world once the linguists are able to decipher the Harappan script as 'inscribed' on the seals, amulets and pottery vessels (Kenoyer, 2003).

A separate section [Section 7, Line: 767] has been added in the revised manuscript.

Major reasons for collapse of Indus valley civilization (IVC):

Many factors - including climatic, economic and political - have been cited in the past as reasons behind decline of IVC. However, no single explanation can be thought of to be the sole descriptor of this decline. These factors perhaps concatenated to eventually led to the fall of IVC.

Climate Change: The dry epoch that lasted for about 900 years due to weakening of Indian Summer monsoon (around 4350 years ago) adversely impacted the agrarian society of IVC (Das, 2018; Dixit et al., 2014). The period of long dry spell reduced the snow cover in northwest Himalaya, causing reduced water availability in Indus river (Dutt et al., 2018; Kathayat et al., 2017). The reduction in water availability severely impacted agricultural systems (Sarkar et al., 2016) and production which ultimately lead to the migration of population towards Gangetic plains.

Infectious Diseases: The vulnerable state of Harappan society is compounded by concurrent social and economic changes, promoting further disintegration of IVC. The stratified social structure and urbanization facilitated propagation of infectious diseases (leprosy, tuberculosis) within the marginalized population. These factors led to massive migration of population from Indus Valley around 1900 B.C. (Schug et al., 2013).

Natural Disasters: The presence of silt deposits, topographic and geological anomalies suggest the occurrence of massive floods was related to the decline of IVC. The tectonic disturbances might have altered the course of Indus river affecting the water availability for agricultural production (Dales, 1966).

*Correspondence to*: P P Mujumdar (pradeep@iisc.ac.in)

**Abstract.** Hydrologic knowledge in India has a historical footprint extending over several millenniums through the Harappan Civilisation (~ 3000 B.C. – 1500 B.C.) and the Vedic Period (~1500-500 B.C.). As in other ancient civilisations across the world, the need to manage water propelled the growth of hydrologic science in ancient India . Most of the ancient hydrologic knowledge, however, has remained hidden and unfamiliar to the world at large until the recent times. In this paper, we provide some fascinating glimpses into the hydrological, hydraulic and related engineering knowledge that existed in ancient India, as discussed in contemporary literature and revealed by the recent explorations and findings. The Vedas, particularly, the *Rigveda*, *Yajurveda* and *Atharvaveda*, have many references to water cycle and associated processes, including water quality, hydraulic machines,  hydro structures and nature-based solutions (NBS) for water management. The Harappan  Civilization epitomizes the level of development of water sciences in ancient India that includes construction of sophisticated hydraulic structures, wastewater disposal systems based on centralized and decentralized concepts as well as methods for wastewater treatments. The Mauryan Empire (~ 322 B.C. – 185 B.C.) is credited as the first "*hydraulic civilization*" characterised by construction of dams with spillways, reservoirs, channels equipped with spillways, Pynes and *Ahars*, understanding of water balance, development of water pricing systems, measurement of rainfall and knowledge of the various hydrological processes. As we investigate deeper into the references of hydrologic works in ancient Indian literature, including the  mythology, many fascinating dimensions of the  Indian scientific contributions emerge. This review presents the various facets of water management exploring  disciplines such as history, archaeology, hydrology and hydraulic engineering,  and  culture, covering the geographical area of the entire Indian sub-continent to the east of the Indus River.  The review covers the period from the Mature Harappan  Civilization to the Vedic Period and the Mauryan Empire.

**1 Introduction**

Water is intimately linked to human existence and is the source of societal and cultural development, traditions, rituals and religious beliefs. The humans created permanent settlements about 10,000 years ago when they adopted an agrarian way of life and began  developing different socio-cultural societies and settlements, largely dependent on water in one way or other (Vuorinen et al., 2007). These developments established a unique relationship between humans and water. Most of the ancient civilizations, e.g., the Indus Valley, Egyptian, Mesopotamian, and Chinese  Civilizations were developed at places where water required for agricultural and human needs was readily available, i.e., in the vicinity of springs, lakes, rivers and low sea levels (Yannopoulos et al., 2015). As water was the prime mover of the ancient civilizations, a clear understanding of the hydrologic cycle, nature and pattern of its various components along with water uses for different purposes led these civilizations to flourish for thousands of years.

The Harappan (or Indus Valley)  Civilization (~3000 B.C. – 1500 B.C.), one of the earliest and most advanced civilizations of the ancient times, was also the world's largest in spatial extent and epitomizes the level of development of science and society in proto-historic Indian sub-continent. The Harappan  Civilization did not hav the 'single state' concept as was practiced by the other contemporary civilizations such as Mesopotamian, pointing to the evidence of centralized control of  palaces, temples and differentiated burials (Kenoyer, 1994; Possehl, 1998, 2003). The Harappan society was based on  shared concepts of power dominance and  patterns of military conquests  has not been found in this society (Kenoyer, 2003). However, more information will be revealed to the world once the linguists  decipher the Harappan script  'inscribed' on the seals, amulets and pottery vessels (Kenoyer, 2003). Jansen (1989) states that the citizens of Harappan Civilization were known for their obsession with water; they prayed to the rivers every day and accorded the rivers a divine status. The urban centres were developed with state-of-the art civil and architectural designs with provisions of sophisticated drainage and waste-water management systems. It is interesting to note in this context that the water and wastewater management systems have been highly amenable to the socio-cultural and socio-economic conditions and religious ways of societies through all the ages of the civilizations (Sorcinelli, 1998; Wolfe, 1999; De Feo and Napoli, 2007; Lofrano and Brown, 2010).

Agriculture was the main economic activity of the Harappan society and an extensive network of reservoirs, wells, canals along with low-cost water harvesting techniques were developed throughout the region at that time (Nair, 2004).  Mohenjo-daro and Dholavira, the two major cities of Indus Valley, are the best examples of advanced water management and drainage systems. The Great Bath of Mohenjo-daro of Indus Valley is considered as the "earliest public water tank of the ancient world" (Mujumdar and Jain, 2018). Adequate archaeological evidence exists to testify that the Harappans of the Indus Valley were well aware of the seasonal rainfall and flooding of the  Indus river during the period between 2500 and 1700 B.C., which is corroborated by modern meteorological investigations (Srinivasan, 1976).

Following the de-urbanization phase (~1900-1500 B.C.) of the Harappan Civilization, the Vedic Period in Indian sub-continent can be bracketed between ~1500-500 B.C. The 'Rigveda' (the earliest of the four Vedas) and many other Vedic texts were composed in this period and in later periods (Kathayat et al., 2017; Witzel, 2014; Sen, 1999). The Vedic Period can be further classified into two stages as the 'Early Vedic Period (~1500-1100 B.C.)' and the 'Late Vedic Period (~1100-500 B.C.)' (Kathayat et al., 2017; Witzel, 1987 & 1999). During the 'Late Vedic Period', the agriculture, metallurgy, commodity production, and trade was largely expanded (Kathayat et al., 2017) and after the 'Late Vedic Period' the period of 'Mahajanpadas' came into existence and which finally converges into the 'Mauryan Empire'. The Vedic texts contain valuable references to 'hydrological cycle'. It was known during Vedic and later times (Rigveda, VIII, 6.19, VIII, 6.20; and VIII, 12.3) (Sarasvati, 2009) that water is not lost in the various processes of hydrological cycle namely evaporation, condensation, rainfall, streamflow, etc., but gets converted from one form to another. At that time Indians were acquainted with cyclonic and orographic effects on rainfall (*Vayu Purana*) and radiation, and convectional heating of earth and evapotranspiration. The Vedic texts and other Mauryan period texts such as '*Arthashastra*' mention about other hydrologic processes such as infiltration, interception, streamflow and geomorphology, including the erosion process. Reference to the hydrologic cycle and artesian wells is available in *Ramayana* (~200 B.C.) (Goswami, 1973). Ground water development and water quality considerations also received sufficient attention in ancient India, as evident from the *Brihat Samhita* (550 A.D.) (Jha, 1988). Topics such as water uptake by plants, evaporation, clouds and their characteristics along with rainfall prediction by observing the natural phenomena of previous years, had been discussed in *Brihat Samhita* (550 A.D.), *Meghamala* (900 A.D.) and other literature from ancient India.

The  *attributed* to Kautilya "who reportedly was the chief minister to the emperor Chandragupta (300 B.C.), the founder of the Mauryan dynasty" (Encyclopaedia Britannica, https://www.britannica.com/topic/Artha-shastra) deals with several issues of governance, including water governance. It mentions about a manually operated cooling device "Variyantra" (revolving water spray for cooling the air). The Variyantra was similar to the water cooler. According to Megasthenes (an ancient Greek historian  who visited the court of King Chandragupta Maurya, around 300 B.C.), the Variyanytra was used by the wealthier sections of the society for cooling the air.  It also gives an extensive account of hydraulic structures built for irrigation and other purposes during the period of the Mauryan Empire (Shamasastry, 1961).

The *Pynes* and *Ahars* (combined irrigation and water management system), reservoir (Sudarshan lake) at Girnar and many other structures were also built during the Mauryan Empire (322-185 B.C.). McClellan III and Dorn, (2015) noted that '… the Mauryan Empire was first and foremost a great *hydraulic  civilization*…'. This  suggests that the technology of the construction of the dams, reservoirs, channels, measurement of rainfall and knowledge of the various hydrological process existed  in the ancient Indian society.  Megasthenes a Greek traveller in Chandragupta's Court, around 300 BC),~~ mentions that

'more than half of the arable land was irrigated  and was in agriculture and produced two harvests in a year'. Further, there was a separate department for supervision, construction and maintenance of a well-developed irrigation system with extensive canals and sluices, wells, lakes and tanks. The same bureau was responsible for planning and settlement of the uncultivated land. A similar description of the different institutional arrangements during Mauryan period can be seen in  *Arthashastra*. The importance of the hydraulic structures in the Mauryan period can be judged on the basis of the punishments/fines imposed on  the offenders. As mentioned in the *Arthashastra*, 'when a person breaks the dam of a tank full of water, he shall be drowned in the very tank; of a tank without water, he shall be punished with the highest amercement; and of a tank which is in ruins owing to neglect, he shall be punished with the middle-most amercement'.

Remarkably, the Mauryan Empire did not lack the other *hallmarks* associated with the hydraulic civilizations (McClellan III and Dorn, 2015). It had the departments concerned with the rivers, excavating and irrigation along with a number of regional and other superintendents such as the superintendent of rivers, agriculture, weights and measures, store-house, space and time, ferries, boats, and ships, towns, pasture grounds, road-cess, and many others along with other  strata of the associated officers such as head of the departments (adhyakshah), collector-general (samahartri), and chamberlain (sannidhatri), etc. Olson (2009) also mentions that there was an extensive irrigation network organised by a state bureaucracy. According to Wittfogel (1955), the Mauryan Empire had virtually all of those characteristics that a hydraulic  Civilization must possess (though it was rather  short lived).

The water pricing was also an important component of the water management system in Mauryan Empire. According to *Arthashastra*, those who cultivate through irrigation (i) by manual labour (*hastaprávartimam*) would have to  pay 1/5th of the produce as water-rate (*udakabhágam*); (ii) by carrying water on shoulders (*skandhaprávartimam*), 1/4th of the produce ; (iii) by water-lifts (*srotoyantraprávartimam*), 1/3rd of the produce; and (iv) by raising water from rivers, lakes, tanks, and wells (*nadisarastatákakúpodghátam*),1/3rd or 1/4th of the produce. The Superintendent of the Agriculture was responsible for compiling the meteorological statistics by using a rain gauge and for observing the sowing of the wet crops, winter crops or summer crops depending on the availability of the water.[2]

~~The Vedic texts, which were composed probably between 1500 and 1200 BC (1700–1100 BC according to some scholars), contain valuable references to 'hydrological cycle'. It was known during Vedic and later times (Rigveda, VIII, 6.19, VIII, 6.20; and VIII, 12.3) (Sarasvati, 2009) that water is not lost in the various processes of hydrological cycle namely evaporation, condensation, rainfall, streamflow, etc., but gets converted from one form to another. Indians were, at that time, acquainted with cyclonic and orographic effects on rainfall (*Vayu Purana*) and radiation, and convectional heating of earth and evapotranspiration. The Vedic texts and other Mauryan period texts such as 'Arthshastra' mention about other hydrologic processes such as infiltration, interception, streamflow and geomorphology, including the erosion process. Reference to the hydrologic cycle and artesian wells is~~

available in *Ramayana* (200 B.C.) (Vālmīki and Goswami, 1973). Ground water development and water quality
considerations also received sufficient attention in ancient India, as evident from the *Brihat Samhita* (550 A.D.)
(Jha, 1988). Topics such as water uptake by plants, evaporation, clouds and their characteristics along with rainfall
prediction by observing the natural phenomena of previous years, had been discussed in *Brihat Samhita* (550
A.D.), *Meghamala* (900 A.D.) and other literature from ancient India.

Historical development of hydro-science has been dealt by many researchers (Baker and Horton, 1936; Biswas,
1969; Chow, 1964). However, not many references to the hydrological contributions in ancient India are found.
Chow (1974) rightly mentions that "… the history of hydrology in Asia is fragmentary at best and much insight
could be obtained by further study". According to Mujumdar and Jain (2018), there is rigorous discussion in
ancient Indian literature on several aspects of hydrologic processes and water resources development and
management practices as we understand them today.

Evidences from ancient water history provide an insight into the hydrological knowledge generated by Indians
more than 3000 years ago. This paper explores the many facets of ancient Indian knowledge on hydrology and
water resources with focus on various hydrological processes, measurement of precipitation, water management
and technology, and wastewater management, based on earlier reviews of the Indian scriptures such as the *Vedas*,
the *ArthasastraArthashastra* (Shamasastry, 1961), *Astadhyayi* (Jigyasu, 1979), *Ramayana* (Vālmīki and
Goswami, 1973), *Mahabharata*, *Puranas*, *Brihat Samhita* (Jha, 1988), *Meghmala*, *Mayurchitraka*, Jain and
Buddhist texts and other ancient texts. In this review, work, we present the state-of-the-art a glimpse of the then
knowledge that existed in ancient India in water sciences, by exploring many disciplines such as history,
archaeology, hydrology and hydraulic engineering, history of technology and history of culture. The paper has
been structured in view offollows the order based on process or technology based order., Wwhile doing so, the
historical order of those processes or technologies has also been followed in each section. The is review work
covering the geographical area of the entire Indian sub-continent to the east of the Indus River. Specifically, it
includes the parts of the Harappan Civilization Civilization(in the present-day Pakistan) and the whole of India
with historical boundaries from the Mature Harappan Civilization Civilization to the Mauryan Empire. These
boundaries encompass the major centres/regions of the development in the ancient India and the 'Mauryan
Empire' has been considered as the a 'logical place' terminal point of the end of the ancient India, which is also
consistent concurrent with the views of Olson (2009) that the 'Mauryan Empire' can be considered as the historical
boundary of the Ancient India.

[revised manuscript text omitted]

Glucklich, (2008) opines about the *Brihat Samhita*: "… as the name of the work itself indicates, its data came from numerous sources, some of them probably quite old. However, the prestige and systematic nature of the *Brihat Samhita* gave its material the authority of prescriptions". Further, it is also appropriate to quote Varahmihira (Chapter 1, Verse, II, *Brihat Samhita*) that '… having correctly examined the substance of the voluminous works of the sages of the past, I attempt to write a clear treatise neither too long nor too short …' (Iyer, 1884). Here, it would be appropriate to recollect words of Murty (1987) that Varahmihira could be considered as the 'earliest hydrologist' of the contemporary world in the same vein as  Leonardo da Vinci being considered the 'Master of Water'.

[revised manuscript text omitted]
, 2010). There is  evidence that the Harappans constructed low cost water harvesting structures such as small check dams, bunds using rock cut pieces and boulders.  The Dholavira city was located between the  ephemeral nallas (streams) Mansar in North and Manhar in South (Figure 4), was equipped with series of small check dams, stone drains for diverting water, bunds to reduce the water velocity and thus reduce siltation in the main reservoirs (Eastern and Western Reservoirs) (Nigam et al., 2016; Agrawal et al., 2018). The Gabarbands were also in use in Harappan civilization. Similarly, the Ahar-Pyne system (an excellent example of Participatory Irrigation Management and Rainwater Harvesting in Mauryan Era) are the examples of low-cost sustainable rainwater harvesting structures. Mohenjo-Daro was one of the major urban centres of the Harappan  Civilization receiving water from at least 700 wells and almost all houses had one private well (Angelakis and Zheng, 2015). The wells were designed as circular to *pipal* (Ficus religiosa) leaf shaped (Khan 2014). Canalising flood waters through ditches for irrigating the Rabi crops (crops of the dry season) was also practiced at that time (Wright, 2010). The farmers of Harappa frequently used "contouring, bunding, terracing, benching, *gabarbands* (dams) and canals for water management (Mckean, 1985). The Gabarbands (stone-built dams for storing and controlling water) were also prevalent in these times for irrigating agricultural lands during the dry seasons (Rabi crops) (Wright, 2010). It may  be noted that the Rabi irrigation was mainly spate irrigation throughout the Indus valley  Civilization (Miller, 2006; Petrie et al., 2017; Petrie, 2019) and water was provided by canals and wells . In the Indus context, it has been argued that perennial and ephemeral water courses were exploited for flood inundation when present, and when not, the inhabitants relied on rainfall, small-scale irrigation, well/lift irrigation and also ponds to supply water (Miller, 2006; Miller, 2015; Petrie, 2019; Weber, 1991, Petrie and Bates, 2017) and  Ahar-Pyne system during the Mauryan era.

During the Vedic age, the principle of collecting water from hilly areas of undulating surface and carrying it through canals to distant areas was known (Bhattacharya, 2012). In the *Rigveda*, many verses indicate that the agriculture can be progressed by use of water from wells, ponds (Verse, I, 23.18 and Verse, V, 32.2). Verse (VIII, 3.10) mentions construction of artificial canals by (Ribhus/Engineer) to irrigate desert areas. Verses (VIII, 49.6 and X, 64.9) emphasizes for efficient use of water, i.e., the water obtained from different sources such as wells, rivers, rain and from any other sources on the earth should be used efficiently, as it is a gift of nature, for well-being of all. There are also references of irrigation by wells (Verse, X. 25), canals (word 'kulya' in *Rigveda*) (Verse, X.99), and digging of the canal (Verse, X75) in the Rigveda. In *Mahâbhâsya* of Patañjali (150 B.C.) the word 'kulya' is also use.

Interestingly, the *Rigveda* (Verses, X 93.12; X 101.7) has a mention of 'asma-cakra' (a wheel made of stones).  water was raised with help of the wheel in a pail using a leather strap. There is also a mention of '*Ghatayantra*' or '*Udghatana*' (a drum-shaped wheel) round which a pair of endless ropes with ghata (i.e. earthen pots) tied at equal distances. In Arabic literature, the water lifting wheel is also known as 'Noria'. Yannopoulos et al., (2015)  state  that the ancient Indians had already developed water lifting and transportation devices. Further, according to Joseph Needham (https://www.machinerylubrication.com/Read/1294/noria-history), based on evidence documented in Indian texts dating from around 350 B.C., the 'Noria' was developed in India around the fifth or fourth century B.C. and the knowledge transmitted to the west by the first century B.C. and to the China by the second century A.D. It is worth mentioning here that during the Vedic period,  water for irrigation purposes was taken from lakes (hrada), canals (kulya), and wells. The exact meaning of the 'asma-cakra' is 'stone-pully' or a 'disk of stone'. The buckets (kosa) tied with the strings made of leather (varatra) were pulled around a stone-pulley and then emptied into the channels (Mukerji, 1960; Yadav, 2008). The *Arthashastra* also mentions irrigating the agricultural fields by raising water from rivers, lakes, tanks and wells using a mechanical device known as 'Udghatam' (Srinivasan, 1970).

[revised manuscript text omitted]

This most likely was the first attempt at treatment on record (Lofrano and Brown, 2010). The pipes were built by well-burned bricks (Gray, 1940) having U-shape cross-section and set in clay mortar with various coverings (brick slabs, flagstones or wooden boards) could be removed easily for cleaning the pipes. These ancient terra-cotta pipes are the precursor of our modern vitrified clay spigot-and-socket sewer pipe (Gray, 1940). These drainage channels were having the provision of cleaning and maintenance by removing the bricks and cut stones (Wolfe, 1999). The cesspits were fitted at the junction of the several drains to avoid the clogging of the drainage systems (Wright, 2010).

Multiple flushing lavatories attached to a sophisticated sewage system were provided in the ancient cities of Harappa and Mohenjo-Daro  Civilization (Pruthi, 2004). The Great Bath at Mohenjo-Daro and the 16 reservoir system of the Dholavira and the Dock yard are the perfect examples of the excellent hydraulic engineering in the Harappan civilization.

Fardin et al., (2013) mention that almost all the settlements of Mohenjo-Daro were connected to the drain network. However, at the same time, at Kalibangan, toilets and bathrooms outflows were connected in U-shaped channels made of wood or terracotta bricks with decentralised sewage systems. These effluents poured into a jar placed in the main street (Chakrabarti, 1995). The same model of wastewater collection was used in Banawali, where effluents were channelled into drains made of clay bricks, before reaching the jars (Bisht, 1984). Several types of stone and terracotta conduits and pipes were also used to transfer water, and drain storm water and wastewater in Minoan  Civilization (ca. 3200–1100 B.C.) (De Feo et al., 2014).

In many other parts of the ancient India, e.g., Jorwe (Maharashtra), a similar drainage system was established during 1375–1050 B.C. (Fardin et al., 2013; Kirk, 1975). Apart from the detailed references on various aspects of hydrology as discussed earlier, we also get some references to water quality in Vedas and other early literature, especially in *Atharvaveda, Charaka Samhita*, and *Susruta Samhita* (both of pre- or early Buddhist era) (NIH, 2018). There are hymns in *Rigveda* stating the role of forest conservation and tree plantation on water quality (Verse V, 83.4). The Verse V, 22.5 of *Atharvaveda*, cautioned people from diseases living in a region with heavy rainfall and bad quality of water. There are instances of classifying water based on taste in epic *Mahabharata* (Verse XII, 184.31 & 224.42). The *Brihat Samhita* also discussed the relationship between soil colour and water quality (Verse, 54.104) and techniques are mentioned for obtaining potable water with medicinal properties from contaminated water (Verses 54.121 & 54.122).

At around 500 B.C., the city of Ujjain was also provided with  sophisticated drainage system having soak-pits built of pottery-ring or pierced pots (Kirk, 1975; Mate, 1969), In Taxila around 300 B.C., very much similar drainage system to that of Mohenjo-Daro was in place (Singh, 2009). This shows that during the ancient times, modern concepts of sanitation and waste water management technology were very well known to the Indians and were in their advanced stages during the Indus Valley Civilization and later periods.

**6. Hydraulic Inter-linkages between the Ancient Indian and Nearby Cultures**

All the ancient civilizations, i.e., Harappan, Egyptian, Mesopotamian, Chinese, and including the Minoan Civilization that flourished and attained their pinnacle were largely dependent on degree/extent of their advancements in water technologies. With  efficient management of water resources, they were able to produce more food grains and mitigate the damages due to natural hazards such as droughts and floods. At the same time, the advanced wastewater management techniques helped in healthy lifestyles, hygiene, and clean environments. The ancient Indian literature covering the period from  the Harappan Civilization to the Vedic Period followed by the Mauryan Empire, and including the hymns and prose in  Vedic Samhitas and Puranas, contains detailed discourses on the various processes of hydrological cycle, including groundwater exploration, water quality, well construction and  irrigation by channels (kulya). Water technological advancements coupled with the architectural sophistication during the Harappan  Civilization were at their zenith. Nowhere in the contemporary world,  such sophisticated and impressive planning relating to the water supply and effluent disposal system could be found (Jansen, 1989). Almost all houses had  private wells with bath and toilet area lined with the standard size burnt bricks and draining into the soak pit or into the street drains.

~~Multiple flushing lavatories attached to a sophisticated sewage system were located in the ancient cities of Harappa and Mohenjo-Daro civilization (Pruthi, 2004). The Great Bath at Mohenjo-Daro and the 16 reservoir system of the Dholavira and the Dock yard are the perfect examples of the excellent hydraulic engineering in the Harappan civilization. The Mauryan Empire was named as the 'hydraulic civilization' due to developments of of the advanced means of irrigation, construction of wells, dams and reservoirs, rainfall measurements, protection of hydraulic structures, and water-pricing systems in place and a stratified establishment of the bureaucratic and engineering establishment.~~

The effluent disposal drainage systems were well-known to almost all the civilizations at that time with varying level of technological advancements. The Egyptian Civilization(~2000-500 B.C.), lacked the flushing lavatories and sophisticated sewer and wastewater disposal systems at that time as was prevalent in Harappan.

The copper pipes were in use in some Pyramids for building bathrooms and sewerage system (De Feo et al., 2014). The Mesopotamian  Civilization(ca. 4000–2500 B.C.) also had well-constructed storm drainage and sanitary sewer systems. However, there seems no system of vertical water supply by means of wells and it was even practically unknown in the early urban cultures (Jansen, 1989; De Feo et al., 2014). According to Jansen (1989) and De Feo et al., (2014), the very efficient drainage and sewerage systems, flushing toilets, which can be compared to the modern ones, re-established in Europe and North America in a century and half ago.

The Mohenjo-Daro city was serviced by at least 700 wells, whereas, the contemporary Egyptian and Mesopotamians had to fetch water bucket-by-bucket from the river and then store in the tanks at homes (Jansen, 1989). The bathing platforms in the Harappan civilizations were also unique as compared to the Mesopotamian and other civilizations. The ancient cities of the Mesopotamian civilization, i.e., UR and Babylone had effective drainage system for storm water control, sewers and drains for household waste and drains specifically for surface runoff (Jones, 1967; Maner, 1966). The ancient Mesopotamians had also developed canal irrigated agriculture and constructed dams across the Tigris river for diverting water to meet the irrigation and domestic supplies. The 'qanat' were widely used in Mesopotamian  Civilization for transferring the water from one place to another using the gravity. The urban centers of the Sumer (Sumerian) and Akkud (Akkadian) (third millennium B.C.) had water supplies by canal(s) connected to the Euphrates River.  The water lifting devices were also used in Mesopotamian  Civilization and the Saaqia (or water wheel) was widely used for lift irrigation using oxen for irrigating the summer crops (Mays, 2008).The '*asma-cakra*' and '*Ghatayantra*' were widely in use during the Vedic and Mauryan Period. The 'Varshaman' was widely used in Mauryan Empire for rainfall measurements. It may be noted that we do not have any reference of 'rainfall measurement' in other contemporary civilizations in the old world. The Pynes-Ahar system of participatory irrigation and rainwater harvesting is a unique system developed in Ancient India. The water-fortification (audaka) around the forts was also a prime requirement in the Mauryan Empire.

In Chinese (Hwang-Ho) civilization, the Shang dynasty (1520-1030 B.C.) developed extensive irrigation works for rice cultivation. Various water works such as dikes, dams, canals and artificial lakes proliferated across the Chinese civilization.  During the period 1100-221 B.C., the Lingzi city (covering an area of 15 km$^2$) also had a complex water supply and drainage system, combined with the river, drainage raceway, pipeline and moat (De Feo, et al., 2014).  Notably,  The underground urban drainage systems were also in existence in Chine during the Shan Dynasty (~10-15 B.C.).

The Minoan  Civilization(~3200-1100 B.C.) is considered to be the first and the most important European culture (Khan et al., 2020). The Crete island was the centre of the Minoan  Civilization and was known for architectural and hydraulic operation of its water supply, sewerage, and drainage systems (Khan et al., 2020). Aqueducts made of terracotta were in use for transporting water from the mountain springs. Water cistern were used for storing rainwater and spring water for further transporting it by using aqueduct. Lavatories with the flushing system were also in use in this civilization.

In words of Jansen (1989), '….for the first time in the history of mankind, the waterworks developed in Harappan  Civilization were to such a perfection which was to remain unsurpassed until the coming of the Romans and the flowering of civil engineering and architecture in classical antiquity, more than 2,000 years later'.

Overall, if we closely look at the scale of the hydro-technologies in all the civilizations, the Harappan  Civilization is not only credited with the more advanced and larger scale application of hydro-technologies (hydrologic, hydraulic and hydro-mechanical) but also worked as a 'landmark' for the contemporary civilizations to achieve the great heights in human civilizations, on the whole.

**7. Decline of Harappan  Civilization– Role of Climate and Natural Disasters**

The decline  of Harappan  Civilization has been still a '' and the topic is still being  'debated in ' the historical and scientific  circles. Many factors such as climatic, economic and political factors have been attributed to the  'spectacular' decline of Harappan civilization.  however, no single explanation can be thought of as the sole descriptor of this decline (Lawler, 2008). Keeping in view the status and developments of the civilization, it is likely that there were multiple factors that went against the sustainability of the Harappan  Civilization and nature related factors are likely to have played a dominant role. Here we list some of the factors which might have eventually led to the decline  of the Harappan civilization.

- Climate Change: The dry epoch that lasted for about 900 years due to weakening of Indian Summer Monsoon (around 4350 years ago) adversely impacted the agrarian society of this  Civilization(Das, 2018; Dixit et al., 2014). The period of long dry spell reduced the snow cover in northwest Himalaya, causing reduced water availability in Indus river (Dutt et al., 2018; Kathayat et al., 2017). The reduction in water availability severely impacted agricultural systems (Sarkar et al., 2016) and production which ultimately led to the migration of population towards Gangetic plains.
- Infectious Diseases: The vulnerable state of Harappan society is compounded by concurrent social and economic changes, promoting further disintegration of Harappans. The stratified social structure and urbanization facilitated propagation of infectious diseases (leprosy, tuberculosis) within the marginalized population. These factors led to massive migration of population from Indus Valley around 1900 B.C. (Schug et al., 2013).
- Natural Disasters: The presence of silt deposits, topographic and geological anomalies suggest the occurrence of massive floods that might have caused  the decline of Harappans. The tectonic disturbances might have altered the course of Indus river affecting the water availability for agricultural production (Dales, 1966).

**6. Summary and Conclusions**

This paper has explored the hydrological developments in ancient India starting from Harappa  Civilization to the Vedic  Period and during the Mauryan Empire using references from Vedas, mythological epics such as *Mahabharata*, *Ramayana*, Jain and Buddhist literature,  the references of *Arthashastra*, *Astadhyayi* and many other Vedic text such as *Puranas* (*Brahmana*, *Linga*, etc.), *Brihat Samhita*, and other ancient literature. The following conclusions can be drawn from this investigation:

1. The Harappa  Civilization epitomizes the level of development in water sciences. Agriculture was the main economic activity of the Harappan society. Extensive network of canals, water storage structures, different types of wells, and low cost and sustainable water harvesting structures were developed during this period. Harappans had created sophisticated water and wastewater management systems, planned network of sewerage systems through underground drains and also had the earliest known system of flush toilets in the world. The Harappa  Civilization is also credited with the first known dockyard in the entire world. The Harappans  were also aware of the oceanic calamities such as Tsunami.

The Vedas, particularly the *Rigveda*, *Atharvaveda* and *Yajurveda* had specifically dwelt upon the hydrologic cycle and various associated processes. The concepts of evaporation, cloud formation, water movement, infiltration and river flow and repetition of cycle are explicitly discussed in these ancient texts. Rigveda also mentions about water lifting device such as *Asma-cakra/Ghatyanta* (similar to Noria), among others. *Ramayana* has also mentioned about hydrologic cycle and artesian wells. *Mahabharata* explains about the monsoon seasons and water uptake process by plants.

2. *Matsya Purana, Vayu Purana*, *Linga Purana*, and *Brahmanda Purana* also mention about the processes of evaporation, formation of clouds due to cyclonic, convectional and orographic effects, rainfall potential of clouds and many other associated hydrological processes.

3. The *Rigveda, Atharvveda, Brihat Samhita, Susrutu Samhita* and *Charaka Samhita* have numerous references of water quality and nature-based solutions (NBS) for obtaining potable water. The Dakargalam Chapter of *Brihat Samhita* dwelt upon the occurrence and distribution of groundwater resources using geographical pointers and soil markers.

4. ~~The Harappa Civilization epitomizes the level of development in water sciences. Extensive network of canals, water storage structures, different types of wells, and low cost and sustainable water harvesting structures were developed during this period. These people had created sophisticated water and wastewater management systems, planned network of sewerage systems through underground drains and also had the earliest known system of flush toilets in the world. The Harappa Civilization is also credited with the first known dockyard in the entire world. Indus people were also aware about the oceanic calamities such as Tsunami.~~

5.4. The first observatory for measuring rainfall using '*Varshamaan*' (raingauge) was established during Mauryan Empire in India. The reservoirs, dams, canals equipped with the spillways were constructed for irrigation and domestic supplies with adequate knowledge of water balance. The water pricing system was developed. Some structures were also constructed considering 50 years' return period. In  water history, the Mauryan period is  recognized as the first and foremost hydraulic civilization. They had also developed a system to forecast rainfall .

5. There are evidences to show that the Harappans had developed one of the smartest urban centres in those ancient times with exemplary fusion of civil, architectural and material sciences. The Indus  Civilization is known to have developed the earliest known systems of flush toilets in the world. They had also developed sophisticated water management systems comprising series of reservoirs, step wells and channels.

6. Agriculture was practised on a large scale having extensive networks of canals for irrigation. The irrigation systems, different types of wells, water storage systems and low cost and sustainable water harvesting techniques were developed throughout the region at that time. There are many evidences that the Harappans constructed low cost water harvesting structures using locally available materials through public participation. Mohenjo-Daro was one of the major urban centres of the Harappan  Civilization receiving water from at least 700 wells and almost all houses had one private well (Angelakis and Zheng, 2015).

7. The Mauryan kings took keen interest in the irrigation schemes. The Ahar-Pyne system of the Mauryan Empire, an excellent example of rainwater harvesting and irrigation management, is still practiced in South Bihar and Chhota Nagpur. A number of hydraulic structures were built during the Mauryan period for irrigation and drinking purposes. An excavation work by Archaeological Survey of India close to Patna revealed a large canal, likely belonging to the Mauryan period which was possibly constructed for navigation and irrigation. Interestingly, a verse of *Atharvaveda* mentions that those who use rainwater by means of rivers, wells, canals for navigation, recreation, agriculture etc., prosper all the time.

8. Tanks (rainwater harvesting structures) were constructed for irrigating the paddy fields in south India about 2000 years ago. The Chola King Karikalan constructed the Grand Anicut on the Cauvery river for flood protection and for irrigation in the Cauvery delta during the 1$^{st}$ century A.D.

9. As early as 2500 B.C., Harappa and Mohenjo-Daro had the world's first urban sanitation systems. The sewage and drainage systems were composed of complex networks, including latrines, soak-pits, cesspools, pipes and channels, connecting the houses.

10. A number of factors might have eventually led to the collapse of the Harappan civilization: a dry epoch that lasted for about 900 years due to weakening of Indian Summer Monsoon; the stratified social structure and urbanization facilitated propagation of infectious diseases; natural disasters including the occurrence of massive floods and tectonic disturbances.

11. The hydrologic knowledge in ancient India was contained in the *shlokas* of scriptures and very few people are conversant with the languages of the scriptures. Hence, the knowledge and wisdom remained largely unknown to the  later generations. Further, the script of the Harappans has not yet been deciphered. If further research is carried out on ancient literature and when the script of the Harappans is deciphered,  many more facts will emerge which may be much more fascinating than
what we know so far.

**Data availability.** No data sets were used in this article.

**Author contributions.** PPM and SKJ conceptualized the paper and its contents. PKS, PD,  SKJ, and PPM
developed the structure of the paper. PKS wrote most parts of the paper;  PD contributed to Section 5. and
also contributed to referencing and formatting the manuscript. SKJ and PPM wrote some parts of the manuscript
as well as reviewed, revised and supervised the progress of manuscript.

**Competing interests.** The authors declare that they have no conflict of interest.

[Figure]

Figure 1: The Symon's raingauge [Source: Raghunath, (2006)].

[Figure]

**Figure 21: Geographical extent of Indus Valley  Civilization[Source: https://commons.wikimedia.org/wiki/File:Indus_Valley_Civilization,_Mature_Phase_(2600-1900_BCE).png].**

[Figure]

**Figure 32: The southern (a) and eastern (b) reservoirs of Dholavira [Source: Iyer, (2019)].**

[Figure]

[Figure]

**Figure 43: Dockyard (a) and ancient Indus port (b) of Lothal [Source: https://www.harappa.com].**

[Figure]

Figure 4: Location of ancient Dholavira City

[Figure]

Latitude: 25.189014
Longitude: 84.210944
Elevation: 90.63m
Accuracy: 3.2m
Time: 23-06-2020 10:52

**Figure 45: Renovated Ahar-Pyne system in Bihar, India.**